



# Global modeling of heterogeneous hydroxymethanesulfonate chemistry

Shaojie Song[1], Tao Ma[2], Yuzhong Zhang[3,4], Lu Shen[1], Pengfei Liu[5], Ke Li[1], Shixian Zhai[1], Haotian Zheng[1,2], Meng Gao[6], Fengkui Duan[2], Kebin He[2], Michael B. McElroy[1]

[1]School of Engineering and Applied Sciences, Harvard University, Cambridge, MA 02138, USA
[2]State Key Joint Laboratory of Environment Simulation and Pollution Control, School of Environment, State Environmental Protection Key Laboratory of Sources and Control of Air Pollution Complex, Beijing Key Laboratory of Indoor Air Quality Evaluation and Control, Tsinghua University, Beijing 100084, China
[3]School of Engineering, Westlake University, Hangzhou 310024, Zhejiang, China
[4]Institute of Advanced Technology, Westlake Institute for Advanced Study, Hangzhou 310024, Zhejiang, China
[5]School of Earth and Atmospheric Sciences, Georgia Institute of Technology, Atlanta, GA 30332, USA
[6]Department of Geography, State Key Laboratory of Environmental and Biological Analysis, Hong Kong Baptist University, Hong Kong SAR, China

Correspondence: Shaojie Song (songs@seas.harvard.edu)

**Abstract.** Hydroxymethanesulfonate (HMS) has recently been identified as an abundant organosulfur compound in aerosols
during winter haze episodes in northern China. It has also been detected in other regions, although the concentrations are low. Because of the sparse field measurements, the global significance of HMS and its spatial and seasonal patterns remain unclear. Here, we implement HMS chemistry into the GEOS-Chem chemical transport model and conduct multiple global simulations. The developed model accounts for cloud entrainment and gas–aqueous mass transfer within the rate expressions for heterogeneous sulfur chemistry. Our simulations can generally reproduce the available HMS observations, and show that East
Asia has the highest HMS concentration, followed by Europe and North America. The simulated HMS shows a seasonal pattern with higher values in the colder period. Photochemical oxidizing capacity affects the competition of formaldehyde with oxidants (such as ozone and hydrogen peroxide) for sulfur dioxide and is a key factor influencing the seasonality of HMS. The highest average HMS concentration (1–3 μg m$^{-3}$) and HMS/sulfate molar ratio (0.1–0.2) are found in northern China winter. The simulations suggest that aqueous clouds act as the major medium for HMS chemistry while aerosol liquid water may play
a role if its rate constant for HMS formation is greatly enhanced compared to cloud water.

## 1 Introduction

Organosulfur compounds (OSs) have been detected in secondary organic aerosols (SOA). The OSs affect the physicochemical properties of aerosols such as hygroscopicity, acidity, and viscosity, and ultimately the climate and health effects of aerosols
(Surratt et al., 2007; Farmer et al., 2010; Sorooshian et al., 2015; Estillore et al., 2016; Riva et al., 2019). The identified OSs include organosulfates (ROSO$_3^-$), sulfoxides (RSOR'), sulfones (RSO$_2$R'), and sulfonates (RSO$_3^-$) (Brüggemann et al., 2020). Sulfonates include methanesulfonate (CH$_3$SO$_3^-$, deprotonated MSA, methanesulfonic acid) and hydroxyalkylsulfonates (RCH(OH)SO$_3^-$) (Song et al., 2019a). These classes of OSs may differ widely in their formation mechanisms, concentration



levels, and spatiotemporal distributions. Organosulfates and MSA are the two most studied OSs species or classes (Bates et al., 1992; Huang et al., 2017; Brüggemann et al., 2020). Organosulfates are primarily formed by the reactive uptake of gas-phase epoxides on acidic sulfate particles (Froyd et al., 2010; Surratt et al., 2010; Xu et al., 2015). The most abundant organosulfate observed in ambient fine particulate matter (PM$_{2.5}$) is the isoprene-derived methyltetrol sulfate (C$_5$H$_{11}$SO$_7^-$), with an average concentration of 1.8 μg m$^{-3}$ found during August 2015 in Atlanta, Georgia, USA (Hettiyadura et al., 2019). MSA is produced primarily by the oxidation of biogenic dimethyl sulfide (DMS) and is likely the major organosulfur species in many regions over the oceans (Chen et al., 2018; Hodshire et al., 2019). The concentrations of aerosol-phase MSA in marine environments are on the order of tens to a few hundreds of ng m$^{-3}$ (Phinney et al., 2006; Sciare et al., 2009; Huang et al., 2017).

Very recently, high mass concentrations of hydroxymethanesulfonate (HMS, CH$_2$(OH)SO$_3^-$), a hydroxyalkylsulfonate species, have been detected in winter Beijing, China using an aerosol mass spectrometer by Song et al. (2019a) and using an improved ion chromatography method by Ma et al. (2020). The mass spectrometry quantification of HMS in ambient aerosols may be subject to the interference of other inorganic and organic sulfur compounds, as suggested by Dovrou et al. (2019). The average HMS concentration in 2015/16 and 2016/17 winters in Beijing was observed to be 1.9 μg m$^{-3}$ (Ma et al., 2020). The highest daily average HMS concentration reached 15 μg m$^{-3}$, accounting for 6% of PM$_{2.5}$ concentration (Ma et al., 2020). Song et al. (2019a) argued that HMS was likely the major organosulfur compound during winter haze events in northern China. Prior to the two studies, only low levels of HMS, with averages on the order of 0.01 μg m$^{-3}$, had been observed in the United States, Japan, and Germany (Dixon and Aasen, 1999; Suzuki et al., 2001; Scheinhardt et al., 2014). Generally, existing observational studies indicate significant spatial variations in HMS.

Our knowledge of the chemical mechanism for HMS stems largely from studies in the 1980s when it was recognized as part of the aqueous sulfur chemistry (Pandis and Seinfeld, 1989). Field measurements of cloud water in the Los Angeles Basin showed the coexistence of H$_2$O$_2$ and S(IV) that was much larger than expected based on the phase equilibrium with gaseous SO$_2$ (Richards et al., 1983). The formation of HMS by the reaction of dissolved SO$_2$ and HCHO was postulated, and then proved, to explain the observed excess of S(IV) (Munger et al., 1986). The laboratory experiments from several groups determined the kinetics and thermodynamics of HMS reactions in aqueous solutions (Boyce and Hoffmann, 1984; Deister et al., 1986; Dong and Dasgupta, 1986; Kok et al., 1986; Olson and Fessenden, 1992). Briefly, both formation and decomposition of HMS depend strongly on pH, i.e., the hydrogen ion activity expressed on a logarithmic scale. HMS is resistant to oxidation by H$_2$O$_2$ and O$_3$ but reacts with hydroxyl radicals (OH) in the aqueous phase. These studies suggested that the atmospheric conditions favorable for the formation and stability of HMS involved abundant gas-phase SO$_2$ and HCHO, high aqueous water content, low temperature, intermediate pH, and low photochemical activity.

The integration and reconciliation of data from field observations, laboratory experiments, and chemical modeling are crucial for obtaining a better understanding of how HMS is processed in the atmosphere. This study offers a global chemical simulation





of HMS using the GEOS-Chem chemical transport model to explore its large-scale spatiotemporal distribution. Multiple model simulations are designed and conducted. The model is driven by the kinetic and thermodynamic data obtained from available laboratory experiments. The simulated results are compared with field observations. The HMS chemistry is heterogeneous in nature since the reactions occur in the aqueous phase with reactants transported from the gas phase (Jacob, 2000). Sometimes

heterogeneous chemistry is referred to as multiphase chemistry (Ravishankara, 1997). Both aqueous cloud droplets (Jacob, 1986; Olson and Hoffmann, 1989; Moch et al., 2018) and aqueous aerosols (Song et al., 2019a; Ma et al., 2020) have been suggested to provide the media for HMS reactions. However, kinetic and thermodynamic data have been determined only in dilute solutions, which are suitable for application in clouds. The lack of corresponding data in concentrated solutions poses a key challenge to modeling HMS chemistry for aerosol water. Therefore, we assume that cloud water serves as the only medium

in the control and default simulations. The role of aerosol water is explored through several sensitivity simulations. As shown in Figure 1, the overall heterogeneous reaction rates are controlled not only by rate constants in the aqueous phase but also by mass transfer limitations between the gas and aqueous phases (Jacob, 1986; Ravishankara, 1997; Seinfeld and Pandis, 2016). In partly cloudy conditions, heterogeneous reactions may also be influenced by the entrainment and detrainment of air into and out from clouds (Fig. 1) (Holmes et al., 2019). Compared with the control simulation that follows the parameterization in

the standard GEOS-Chem model, the default simulation improves treatments of entrainment and mass transfer processes for heterogeneous cloud sulfur chemistry. Aerosol water chemistry in the sensitivity simulations also considers the physiochemical processes in Fig. 1, allowing an evaluation of the importance of the two aqueous media.

This article is organized as follows. In the Method section, we first provide an overview of the aqueous chemical reactions for

HMS, including its formation, decomposition, and oxidation (Sect. 2.1). From existing laboratory studies, we critically estimate the best values and uncertainties of their rate constants. The general configuration of the GEOS-Chem model is described in Sect. 2.2, including its version, simulation period, spatial and temporal resolutions, meteorological field, chemical mechanisms, and underlying emissions. A brief introduction of sulfur simulation in the standard model is given in Sect. 2.3. The two major simulations in this study, control and default, are described in Sect. 2.4 and 2.5, respectively. Based on settings in the standard

model, the control simulation implements heterogeneous HMS chemistry using cloud as the only aqueous medium. We find that the in-cloud $SO_2$ titration by various reactants is inappropriately represented in the control simulation, very likely leading to an overestimation of HMS formation. The default simulation fixes this issue. Sect. 2.6 describes the sensitivity simulations designed to investigate the key factors leading to uncertainty in the modeled HMS levels. In the Results and discussion section, we first show in Sect. 3.1 the spatial and seasonal distributions of HMS from the default simulation and discuss the underlying

factors. Differences in the modeled HMS between the default and control simulations are presented and discussed in Sect. 3.2. Sect. 3.3 demonstrates the key uncertain parameters and processes in the HMS model identified from sensitivity simulations. Sect. 3.4 compares the observations of HMS in different regions with model results. The knowledge gained in this study and the remaining gaps are summarized in Sect. 3.5. Finally, the conclusions are given in Sect. 4.





## 2 Methods

### 2.1 Kinetics and thermodynamics of HMS chemistry

Hydroxymethanesulfonic acid (HMSA, $CH_2(OH)SO_3H$) is a diacid with $pK_{a1} < 0$ (R1) and $pK_{a2} \sim 12$ (R2). Thus, it primarily exists as HMS ($CH_2(OH)SO_3^-$) in tropospheric clouds and aerosols. In the aqueous phase, HMS is produced by the nucleophilic

addition of $HSO_3^-$ and $SO_3^{2-}$ to the carbonyl C atom of HCHO (R3–R6). As $SO_3^{2-}$ is a much stronger nucleophile than $HSO_3^-$, the rate constant of $HCHO_{(aq)} + SO_3^{2-}$, $k_2$, is a few orders of magnitude higher than that of $HCHO_{(aq)} + HSO_3^-$, $k_1$, as shown in Table 1. $HCHO_{(aq)}$ refers to the free, unhydrated formaldehyde dissolved in the aqueous phase, and maintains an equilibrium with its hydrated form, $CH_2(OH)_2$ (methylene glycol). The equilibrium constant of (R7), $K_h$, represents the extent of hydration (Eq. 1). (R1–R6) are all reversible and can be summarized by (R8). $SO_{2(aq)}^T$ is the sum of $SO_2 \cdot H_2O$, $HSO_3^-$, and $SO_3^{2-}$ (Eq. 2).

$k_f$ ($M^{-1}s^{-1}$) and $k_d$ ($s^{-1}$) represent the forward and backward reaction (HMS formation and decomposition) rate constants of (R8) and $K_{eq}$ ($M^{-1}$) is its equilibrium constant (Eq. 3). $k_f$ is a combination of $k_1$ and $k_2$ weighted by the fractions of $HSO_3^-$ and $SO_3^{2-}$ in $SO_{2(aq)}^T$ (Eqs. 4–6). $K_{s1}$ and $K_{s2}$ denote the first and second dissociation constants for dissolved $SO_2$ (Table 2). Figure 2 shows the values of $k_f$ and $k_d$ obtained from the available laboratory experiments as a function of pH (Blackadder and Hinshelwood, 1958; Sørensen and Andersen, 1970; Boyce and Hoffmann, 1984; Deister et al., 1986; Dong and Dasgupta,

1986; Kok et al., 1986; Lagrange et al., 1999). In general, we find a large discrepancy for $k_f$ and good agreement for $k_d$.

$$CH_2(OH)SO_3H \leftrightarrow CH_2(OH)SO_3^- + H^+ \tag{R1}$$

$$CH_2(OH)SO_3^- \leftrightarrow CH_2(O^-)SO_3^- + H^+ \tag{R2}$$

$$SO_2 \cdot H_2O \leftrightarrow HSO_3^- + H^+ \tag{R3}$$

$$HSO_3^- \leftrightarrow SO_3^{2-} + H^+ \tag{R4}$$

$$HCHO_{(aq)} + HSO_3^- \overset{k_1}{\leftrightarrow} CH_2(OH)SO_3^- \tag{R5}$$

$$HCHO_{(aq)} + SO_3^{2-} \overset{k_2}{\leftrightarrow} CH_2(O^-)SO_3^- \tag{R6}$$

$$HCHO_{(aq)} + H_2O \leftrightarrow CH_2(OH)_2 \tag{R7}$$

$$HCHO_{(aq)} + SO_{2(aq)}^T \leftrightarrow HMS \tag{R8}$$

$$K_h = [CH_2(OH)_2]/[HCHO]_{aq} \tag{1}$$

$$\left[SO_2^T\right]_{aq} = [SO_2 \cdot H_2O] + [HSO_3^-] + \left[SO_3^{2-}\right] \tag{2}$$

$$K_{eq} = [HMS]/\left([HCHO]_{aq}\left[SO_2^T\right]_{aq}\right) = k_f/k_d \tag{3}$$

$$k_f = k_{10}x_{HSO_3^-} + k_{11}x_{SO_3^{2-}} \tag{4}$$

$$x_{HSO_3^-} = [HSO_3^-]/\left[SO_2^T\right]_{aq} = K_{s1}[H^+]/([H^+]^2 + K_{s1}[H^+] + K_{s1}K_{s2}) \tag{5}$$

$$x_{SO_3^{2-}} = [SO_3^{2-}]/\left[SO_2^T\right]_{aq} = K_{s1}K_{s2}/([H^+]^2 + K_{s1}[H^+] + K_{s1}K_{s2}) \tag{6}$$



### 2.1.1 HMS formation

Boyce and Hoffmann (1984) determined the following kinetic parameters at ionic strength $\mu = 1$ M, pH from 0 to 3.5: $k_1 = 7.9 \times 10^2$ M$^{-1}$ s$^{-1}$ and $k_2 = 2.5 \times 10^7$ M$^{-1}$ s$^{-1}$ (both at 298 K). The enthalpies of activation $\Delta^{\ddagger}H_1$ and $\Delta^{\ddagger}H_2$ were 25 kJ mol$^{-1}$ and 20 kJ mol$^{-1}$, respectively. These parameters were calculated assuming $K_{s1} = 1.45 \times 10^{-2}$ M and $K_{s2} = 6.31 \times 10^{-8}$ M,

which were in fact for dilute solutions ($\mu \approx 0$ M). According to Boyce and Hoffmann (1984), application of the Davies equation to correct for the ionic strength effects on $K_{s1}$ and $K_{s2}$ yielded $k_1 = 4.5 \times 10^2$ M$^{-1}$s$^{-1}$, $k_2 = 5.4 \times 10^6$ M$^{-1}$s$^{-1}$ (both at 298 K), $\Delta^{\ddagger}H_1 = 22$ kJ mol$^{-1}$, and $\Delta^{\ddagger}H_2 = 21$ kJ mol$^{-1}$. Boyce and Hoffmann (1984) also used a higher $K_h$ of $1.8 \times 10^3$ than the value of $1.3 \times 10^3$ obtained in a more recent study by Winkelman et al. (2002) (Table 2). We further adjust the kinetics based on this recent $K_h$ and obtain $k_1 = 3.2 \times 10^2$ M$^{-1}$s$^{-1}$ and $k_2 = 3.8 \times 10^6$ M$^{-1}$s$^{-1}$.

Therefore, two sets of HMS formation kinetic data can be obtained from Boyce and Hoffmann (1984) and are designated here as the high and low rate constants, as shown in Table 1 and Fig. 2. The calculated high and low $k_f$ differ by a factor of about 3 at pH < 2 and by a factor of about 6 at pH > 4. The low $k_f$ agrees very well (within a factor of 1.1) with the results determined by Kok et al. (1986) and Deister et al. (1986) at higher pH 4, 5, and 5.6 (Fig. 2). The low kinetic data are also closer to the rate

constants from the recent quantum chemical calculations by Zhang et al. (2019) ($k_1 = 0.9$ M$^{-1}$s$^{-1}$, $k_2 = 2 \times 10^6$ M$^{-1}$s$^{-1}$, at 298 K). Consequently, the low formation rate constants from Boyce and Hoffmann (1984) are adopted for the default model simulation, while the high ones are used in a sensitivity simulation. Lagrange et al. (1999) proposed another value of $k_f$ which was about 1–4 orders of magnitude smaller than the low $k_f$ from Boyce and Hoffmann (1984) at pH > 4 (Fig. 2). The simulated HMS concentration is negligible everywhere when applying the $k_f$ from Lagrange et al. (1999) in the model, and thus, will not

be discussed further.

### 2.1.2 HMS decomposition

The most complete analysis of $K_{eq}$ was done by Deister et al. (1986). We calculate the expression of $k_d$ using the low $k_f$ from Boyce and Hoffmann (1984) and $K_{eq}$ from Deister et al. (1986) (Eq. 3 and Table 1). As shown in Fig. 2, $k_d$ estimated in this way agrees within a factor of about 2 with results from the other laboratory studies (Blackadder and Hinshelwood, 1958;

Sørensen and Andersen, 1970; Dong and Dasgupta, 1986; Kok et al., 1986; Lagrange et al., 1999). Therefore, this expression of $k_d$ is adopted in the default simulation, and its value is doubled in a sensitivity simulation. If we use the high $k_f$ from Boyce and Hoffmann (1984) and the $K_{eq}$ from Deister et al. (1986), we will obtain a $k_d$ that is several times higher than estimates from the other studies. This may serve as circumstantial evidence in favor of the low $k_f$.

### 2.1.3 HMS oxidation

HMS is resistant to oxidation by $H_2O_2$ and $O_3$ but can be oxidized by OH in the aqueous phase (Martin et al., 1989; Olson and Fessenden, 1992). (R9) produces HCHO and peroxysulfate radical (SO$_5^-$) with a rate constant of $2.7 \times 10^8$ M$^{-1}$ s$^{-1}$ (Olson and





Fessenden, 1992) (Table 1). This value is lower by a factor of about 4 than the results reported in two earlier laboratory studies (Martin et al., 1989; Deister et al., 1990). Olson and Fessenden (1992) argued that these two studies were subject to artifacts and interferences from secondary reactions.

$$\text{HMS} + \text{OH}_{(aq)} \xrightarrow{\text{O}_2} \text{HCHO}_{(aq)} + \text{SO}_5^- + \text{H}_2\text{O} \tag{R9}$$

The second source of uncertainty in (R9) arises from estimating aqueous OH concentrations. Aqueous OH is a short-lived species that can be transferred from the gas phase and generated/scavenged in the aqueous phase. Its sources and sinks, which are linked to photochemical processes (e.g., photolysis of nitrate and peroxides), transition metal ions (Fenton reactions), and/or reactions with halogen anions and organic matters, are not yet fully understood (Tilgner and Herrmann, 2018). Currently, there exist significant discrepancies between the modeled and measured $[\text{OH}]_{aq}$ levels. A comprehensive overview has shown that $[\text{OH}]_{aq}$ from different model studies ranges from $3 \times 10^{-15}$ M to $8 \times 10^{-12}$ M for cloud droplets and from $1 \times 10^{-16}$ M to $8 \times 10^{-12}$ M for aqueous aerosols, and that, on the other hand, data ranges of the measured $[\text{OH}]_{aq}$ are $0.5$–$7 \times 10^{-15}$ M for clouds and $0.1$–$6 \times 10^{-15}$ M for aerosols (Tilgner and Herrmann, 2018). On average, the modeled $[\text{OH}]_{aq}$ is two orders of magnitude higher than the measured values. This large gap is believed to result from the limitations of both models and measurements. The bulk measurements of $[\text{OH}]_{aq}$ may underestimate its concentrations in real aerosols and clouds due to lack of replenishment of important oxidations and OH precursors from the gas phase under the dark conditions of sample storage and treatment. On the other hand, the multiphase models may significantly overpredict $[\text{OH}]_{aq}$ because they only partially consider the complex organic aqueous chemistry. The reasonable estimates of $[\text{OH}]_{aq}$ in real aerosols and clouds seem to be one order of magnitude lower than modeled concentrations and one order of magnitude higher than measured levels (Tilgner and Herrmann, 2018). Since GEOS-Chem does not have a detailed representation of aqueous OH chemistry, we simply estimate $[\text{OH}]_{aq}$ using the modeled $[\text{OH}]_g$ and a pseudo Henry's law constant $H_{\text{OH}}^*$ (Eq. 7). In the default simulation, $H_{\text{OH}}^*$ is set to $4 \times 10^{-20}$ M cm³ molecules⁻¹. $H_{\text{OH}}^*$ is more than one order of magnitude lower than its intrinsic Henry's law constant, $H_{\text{OH}}$ (Table 2), reflecting our presumption that the various organic and inorganic compounds in the aqueous phase act as a net sink for OH radicals. A global mean $[\text{OH}]_g$ of about $1 \times 10^6$ molecules cm⁻³ implies a mean $[\text{OH}]_{aq}$ of $4 \times 10^{-14}$ M, one order of magnitude higher than the mean of the above-mentioned measured $[\text{OH}]_{aq}$.

$$[\text{OH}]_{aq} = [\text{OH}]_g \times H_{\text{OH}}^* \tag{7}$$

The products of (R9) are $\text{HCHO}_{(aq)}$ and $\text{SO}_5^-$. Interestingly, the net effect of HMS formation (R8) and its subsequent oxidation (R9) is the oxidation of $\text{SO}_{2(aq)}^T$ by $\text{OH}_{(aq)}$, which represents thus an indirect oxidation pathway for SO₂. The sinks for $\text{SO}_5^-$ are mainly the reactions with $\text{O}_2^-$, $\text{HCOO}^-$, and itself (R10–R12). The reaction of $\text{SO}_5^-$ and $\text{HSO}_3^-$ is slow (Jacob et al., 1989). The peroxymonosulfate radical ($\text{HSO}_5^-$) produced by (R10–R11) can oxidize $\text{HSO}_3^-$ to sulfate (R13) with a similar rate constant to $\text{H}_2\text{O}_2 + \text{HSO}_3^-$ (Betterton and Hoffmann, 1988). The sulfate radical ($\text{SO}_4^-$) produced by (R12) is a very strong oxidant and can



react rapidly with $HSO_3^-$ and $SO_3^{2-}$ (R14–R15) as well as with many other species such as $Cl^-$, $NO_2^-$, $O_2^-$, $HCOO^-$, and $HO_2$ (Jacob, 1986). The rate constants for (R10–R15) can be found in Jacob et al. (1989). It is convenient to define the sulfate yield as the number of $SO_4^{2-}$ ions produced due to each attack of $OH_{(aq)}$ on HMS. If $SO_5^-$ reacts with $O_2^-/HCOO^-$ (R10–R11) and the product $HSO_5^-$ oxidizes $HSO_3^-$ (R13), the yield is 2. If $SO_5^-$ undergoes self-reaction (R12) and the produced $SO_4^-$ reacts with $HSO_3^-/SO_3^{2-}$ (R14–R15), a reaction chain is triggered as the products include $SO_5^-$. In certain conditions, the sulfate yield can reach several tens or more (Jacob et al., 1989). However, as mentioned above, other oxidizable species also compete for $SO_4^-$, thereby terminating this chain and leading to a sulfate yield of 1. In remote environments where $SO_2$ is very low, $HSO_5^-$ may be a stable species, resulting in a sulfate yield < 1. Our low $[OH]_{aq}$ assumption implies the existence of important oxidizable species, and therefore, the chain propagation is limited. In our simulations, the sulfate yield is assumed to be 2.

$$SO_5^- + O_2^- \xrightarrow{H_2O} HSO_5^- + O_2 + OH^- \tag{R10}$$

$$SO_5^- + HCOO^- \xrightarrow{O_2} HSO_5^- + O_2^- + CO_2 \tag{R11}$$

$$SO_5^- + SO_5^- \rightarrow 2SO_4^- + O_2 \tag{R12}$$

$$HSO_5^- + HSO_3^- \rightarrow 2SO_4^{2-} + 2H^+ \tag{R13}$$

$$SO_4^- + HSO_3^- \xrightarrow{O_2} SO_4^{2-} + SO_5^- + H^+ \tag{R14}$$

$$SO_4^- + SO_3^{2-} \xrightarrow{O_2} SO_4^{2-} + SO_5^- \tag{R15}$$

### 2.1.4 Phase equilibrium

The gas/aqueous phase equilibriums of HCHO (R16) and $SO_2$ (R17) are described by intrinsic Henry's law constants, $H_{HCHO}$ and $H_{SO_2}$, respectively (Table 2). $HCHO_{(aq)}$ is subject to hydration, and the apparent Henry's law constant, $H^*_{HCHO}$, is much larger than $H_{HCHO}$ (Eq. 8). $SO_2 \cdot H_2O$ dissociates twice in the aqueous phase and thus $H^*_{SO_2}$ depends on pH (Eq. 9). The rates for the hydration of $HCHO_{(aq)}$ ($k_h$ in Table 2) and the acid dissociations of $SO_2 \cdot H_2O$ (Schwartz and Freiberg, 1981) are fast enough and we assume that these reactions are always in equilibrium.

$$HCHO_{(g)} \leftrightarrow HCHO_{(aq)} \tag{R16}$$

$$SO_{2(g)} + H_2O \leftrightarrow SO_2 \cdot H_2O \tag{R17}$$

$$H^*_{HCHO} = \left([CH_2(OH)_2] + [HCHO]_{aq}\right)/[HCHO]_g = H_{HCHO}(1 + K_h) \cong H_{HCHO}K_h \tag{8}$$

$$H^*_{SO_2} = \left[SO_2^T\right]_{aq}/[SO_2]_g = H_{SO2}(1 + K_{s1}/[H^+] + K_{s1}K_{s2}/[H^+]^2) \tag{9}$$

### 2.2 General model description

We perform global simulations of heterogeneous HMS chemistry using the three-dimensional GEOS-Chem chemical transport model (version 12.1.0, Doi: 10.5281/zenodo.1553349, last access: 10 June 2020). The simulations are driven by the MERRA-2 (Modern-Era Retrospective analysis for Research and Applications, version 2) reanalysis meteorology from the NASA Goddard Earth Observing System (Gelaro et al., 2017). The original MERRA-2 has a resolution of 0.625° (longitude) × 0.5°


(latitude) and is degraded to 5° × 4° for input into the simulations. There are 47 vertical layers in the atmosphere from surface to the mesosphere. The simulations are conducted for 18 months starting from March 2015. The first 6 months are used for initialization and we focus on the 1-year simulation results from September 2015 to August 2016. These months are selected to obtain a continuous boreal winter. We use the tropospheric chemistry mechanism with detailed reactions for $O_3$-$NO_x$-VOC

(volatile organic compound)-aerosol-halogen interactions. The time step for species advection, vertical mixing, and convection is set to 10 min. The time step is 20 min for emissions, dry deposition, photolysis, and chemistry, as recommended by Philip et al. (2016). The simulated aerosol species include secondary inorganic (sulfate, nitrate, and ammonium) and organic aerosols, primary organic aerosols, black carbon, dust, and sea salt.

Emissions are calculated using HEMCO (the Harvard-NASA Emissions Component, version v2.1.010) (Keller et al., 2014). The global anthropogenic emissions of $SO_2$, $NO_x$, $NH_3$, CO, VOCs, black carbon, and organic carbon are from the Community Emissions Data System (CEDS) (Hoesly et al., 2018). Emissions are overwritten by regional inventories wherever available: EMEP (European Monitoring and Evaluation Programme) over Europe (www.emep.int/index.html, last access: 10 June 2020), MIX over Asia (Li et al., 2017), DICE (Diffuse and Inefficient Combustion Emissions) over Africa (Marais and Wiedinmyer,

2016), NEI (National Emissions Inventory) over the United States (Travis et al., 2016), CAC (Criteria Air Contaminants) over Canada (wiki.seas.harvard.edu/geos-chem/index.php/CAC_anthropogenic_emissions, last access: 10 June 2020), and MEIC (Multi-resolution Emission Inventory) over China (Zheng et al., 2018). Primary emissions of sulfate constitute 1.4%–5% of total anthropogenic sulfur emissions in different regions of the world. Aircraft emissions are from the Aviation Emissions Inventory Code (Simone et al., 2013). Biomass burning emissions are from the Global Fire Emissions Database (GFED,

version 4) (van der Werf et al., 2017). Biogenic VOC emissions are calculated by the Model of Emissions of Gases and Aerosols from Nature (MEGAN, version 2.1) (Guenther et al., 2012). Mineral dust emissions follow Duncan et al. (2007) and are distributed in one fine and three coarse size bins. Anthropogenic emissions of fine dust aerosols are from the Anthropogenic Fugitive, Combustion, and Industrial Dust (AFCID) inventory (Philip et al., 2017). Sea salt aerosols in two size bins (fine and coarse) are simulated based on Jaeglé et al. (2011). Other emissions include volcanic $SO_2$ emissions (Ge et al., 2016), oceanic

DMS emissions (Lana et al., 2011), lightning and soil $NO_x$ emissions (Hudman et al., 2012; Murray et al., 2012), and natural $NH_3$ emissions from the GEIA (Global Emissions InitiAtive) inventory (www.geiacenter.org, last access: 10 June 2020).

Because of the importance of acidity for heterogeneous HMS chemistry, more details are provided for the calculation of cloud water and aerosol pH. The standard model calculates cloud water pH iteratively with an initial estimate of 4.5, as described in

Alexander et al. (2012). The ions considered in the electroneutrality equation are $NH_4^+$, $H^+$, $OH^-$, $SO_4^{2-}$, $NO_3^-$, $HSO_3^-$, $SO_3^{2-}$, $HCO_3^-$, and $CO_3^{2-}$. $HSO_3^-$/$SO_3^{2-}$ and $HCO_3^-$/$CO_3^{2-}$ are from the scavenging of $SO_2$ and $CO_2$. $SO_4^{2-}$ is assumed to be the only form of sulfate and is obtained from the cloud scavenging of aerosols. $NH_4^+$ and $NO_3^-$ are from the scavenging of both aerosols and gases ($NH_3$ and $HNO_3$). The scavenging efficiencies of aerosols and gases are assumed to be 0.7 and unity, respectively. The ISORROPIA II (version 2.2) thermodynamic equilibrium model (Fountoukis and Nenes, 2007) is used to calculate the





inorganic aerosol water content (m$^3$ H$_2$O m$^{-3}$ air) and pH, including the following gas and aerosol species: NH$_3$, HNO$_3$, ammonium, nitrate, sulfate, and fine sea-salt aerosols.

## 2.3 Sulfur simulation in the standard model

The sulfur simulation in GEOS-Chem has been developed and improved based on multiple studies (Chin et al., 2000; Park et al., 2004; Alexander et al., 2005, 2009; Chen et al., 2017; Shao et al., 2019). The simulated sulfur species include DMS, SO$_2$, MSA, and sulfate. It includes primary emissions of DMS, SO$_2$, and sulfate (Sect. 2.2). SO$_2$, MSA, and sulfate can be formed also by chemical reactions. The model contains three gas-phase reactions of DMS oxidation, producing SO$_2$ and MSA (R18–R20). An expanded chemistry mechanism for DMS can be found in Chen et al. (2018). The oxidation of SO$_2$ to sulfate occurs in the gas phase by OH (R21) and in the aqueous clouds. The aqueous-phase oxidants are O$_3$, H$_2$O$_2$, O$_2$ (catalyzed by transition metal ions Mn$^{2+}$ and Fe$^{3+}$), and HOBr (R22–R25). The effect of the heterogeneity in cloud droplet pH on sulfate production rates is accounted for using the parameterization by Yuen et al. (1996) and Fahey and Pandis (2001). This parameterization is restricted over the ocean since the heterogeneity in pH is believed to be caused by alkaline sea-salt aerosols (Alexander et al., 2012). The model also includes the oxidation of SO$_2$ by O$_3$ on sea-salt aerosol surface (R26) (Alexander et al., 2005).

$$DMS_{(g)} + OH_{(g)} \rightarrow SO_{2(g)} + CH_3O_{2(g)} + HCHO_{(g)} \tag{R18}$$

$$DMS_{(g)} + OH_{(g)} \rightarrow 0.75SO_{2(g)} + 0.25MSA_{(g)} \tag{R19}$$

$$DMS_{(g)} + NO_{3(g)} \rightarrow SO_{2(g)} + CH_3O_{2(g)} + HCHO_{(g)} + HNO_{3(g)} \tag{R20}$$

$$SO_{2(g)} + OH_{(g)} \xrightarrow{M} H_2SO_{4(g)} + HO_{2(g)} \tag{R21}$$

$$SO_{2(aq)}^{T} + O_{3(aq)} \rightarrow SO_4^{2-} + O_{2(aq)} \tag{R22}$$

$$SO_{2(aq)}^{T} + H_2O_{2(aq)} \rightarrow SO_4^{2-} + H_2O \tag{R23}$$

$$SO_{2(aq)}^{T} + O_{2(aq)} \xrightarrow{Mn^{2+},Fe^{3+}} SO_4^{2-} \tag{R24}$$

$$SO_{2(aq)}^{T} + HOBr_{(aq)} \rightarrow SO_4^{2-} + HBr_{(aq)} \tag{R25}$$

$$SO_{2(g)} + O_{3(g)} + \text{fine sea salt} \rightarrow SO_4^{2-} + O_2 \tag{R26}$$

## 2.4 Control simulation

Based on the standard model v12.1.0, we implement heterogeneous HMS chemistry and assume that cloud water provides the only aqueous medium. As described in Sect. 2.1, HMS is produced by dissolved SO$_2$ and HCHO, undergoes decomposition, and is oxidized to sulfate by aqueous OH. Two other cloud sulfate formation pathways are also incorporated, in which SO$_2$ is oxidized by NO$_2$ and HONO (R27–R28).

$$SO_{2(aq)}^{T} + 2NO_{2(aq)} \rightarrow SO_4^{2-} + 2HONO_{(aq)} \tag{R27}$$

$$SO_{2(aq)}^{T} + HONO_{(aq)}^{T} \rightarrow SO_4^{2-} + 0.5N_2O_{(aq)} \tag{R28}$$





Tables 1 and 2 show all the aqueous-phase reaction rate constants and the reactants' Henry's law constants. The solubilities of transition metals Fe and Mn are reduced following Shao et al. (2019). Ten advected tracers are added: one is the aerosol HMS species and the others represent different sulfate formation pathways. Transport and deposition of these tracers are treated in the same way as the sulfate tracer. In addition, several other changes are made in the control simulation to the standard model.

First, we update the dry deposition scheme and the reactive uptake coefficients of $NO_2$, $NO_3$, and $N_2O_5$ on aerosols, following Jaeglé et al. (2018) and Shah et al. (2018). Second, this simulation includes some updates developed by Luo et al. (2019, 2020) in the treatments of wet processes, allowing for spatially and temporally varying in-cloud condensation water contents, empirical washout rates for water-soluble aerosols and nitric acid, the cloud fraction available for aqueous chemistry, and rainout efficiencies for water-soluble aerosols and gases. Third, more ions are included in the cloud water pH calculation. We

consider $Ca^{2+}$, $Mg^{2+}$, $NH_4^+$, $Na^+$, $H^+$, $OH^-$, $Cl^-$, $SO_4^{2-}$, $NO_3^-$, $NO_2^-$, $HSO_3^-$, $SO_3^{2-}$, $HCO_3^-$, $CO_3^{2-}$, $HCOO^-$, $CH_3COO^-$, HMS, and $CH_3SO_3^-$. The Newton-Raphson method is used to find the solution to the cubic electroneutrality equation (Luo et al., 2020). $Ca^{2+}$ and $Mg^{2+}$ are assumed to constitute 3% and 0.6%, respectively, of the dust by mass (Claquin et al., 1999; Fairlie et al., 2010; Nickovic et al., 2012; Shao et al., 2019). Only $Na^+$ and $Cl^-$ from sea-salt aerosols are considered. HMS and $CH_3SO_3^-$ are from the cloud scavenging of aerosols. $NO_2^-$, $HCOO^-$, and $CH_3COO^-$ are from the scavenging of HONO, HCOOH, and $CH_3COOH$, respectively. Fourth, HMS, $CH_3SO_3^-$, and $Ca^{2+}$ and $Mg^{2+}$ in fine dust are included in the

ISORROPIA calculations. We assume the same hygroscopicity of HMS and MSA as sulfate (Xu et al., 2020).

We evaluate the performance of the control simulation by comparing it with the standard GEOS-Chem v12.1.0 (GC12.1.0). Figure S1 shows the horizontal distributions of surface $SO_4^{2-}$ and $SO_2$ concentrations. The global average $SO_4^{2-}$ in the control

simulation is reduced by 24% compared to GC12.1.0. The updates in the treatments of wet processes by Luo et al. (2019; 2020) are primarily responsible for this difference. The $SO_4^{2-}$ concentrations modeled in the control simulation are consistent with the improved model results in Luo et al. (2020), which have been found to agree well with $SO_4^{2-}$ observed in the United States, Europe, and Asia (Luo et al., 2020). Moreover, since GC12.1.0 was released in late 2018, it is necessary to compare it with a more recent model version. Accordingly, we conduct a simulation using the standard GEOS-Chem v12.7.0 (GC12.7.0, released

in February 2020, wiki.seas.harvard.edu/geos-chem/index.php/GEOS-Chem_12, last access: 10 June 2020). We find that the global average $SO_4^{2-}$ in GC12.7.0 only differs little (3%) compared with that in GC12.1.0 (Fig. S2).

Below, we provide details on the calculation of cloud sulfur chemistry and highlight the need for more accurate representations of in-cloud $SO_2$ titration by various reactants, which include $O_3$ (R22), $H_2O_2$ (R23), $O_2$ (R24), HOBr (R25), $NO_2$ (R27), HONO

(R28), and HCHO (R8). Cloud sulfur chemistry is calculated locally in the model grid cells where aqueous clouds are present. $f_c$ (dimensionless, $0 \leq f_c \leq 1$) denotes the fraction of aqueous cloud in a grid cell, and $L$ ($m^3$ $H_2O$ $m^{-3}$ air) denotes the in-cloud liquid water content. In each chemistry time step ($\Delta t = 20$ min), the losses of $SO_2$ in the above reactions (R8, R22–R25, R27–R28) are calculated. R24 is treated as a first-order reaction of $SO_2$ ($O_2$ is in large excess), while the other reactions are second





order. The first- and second-order rate constants for the aqueous reaction of $SO_{2(aq)}^T$ and $X_{i(aq)}$, $k_{1,aq,i}$ (s$^{-1}$) and $k_{2,aq,i}$ (M$^{-1}$ s$^{-1}$), are obtained by Eq. 10 from the kinetic data in Table 1. $X_i$ ($i = 1:7$) represents the $i^{th}$ reactant with $SO_2$. $R_{aq,i}$ is the reaction rate (M s$^{-1}$). $k_{1,aq,i}$ and $k_{2,aq,i}$ are used to derive the first- and second-order rate constants for the heterogeneous reaction of $SO_{2(g)}$ and $X_{i(g)}$, $k_{1,g,i}$ (s$^{-1}$) and $k_{2,g,i}$ (mol mol$^{-1}$ s$^{-1}$) (Eq. 11). $H_{SO2}^*$ and $H_{Xi}$ indicate their Henry's law constants. $f_{g,SO2}$ and $f_{g,Xi}$ are

the gas-phase partitioning fractions of $SO_2$ and $X_i$, respectively (Eq. 12). $R$ is the gas constant. $T$ (K) is the temperature. $P$ (atm) is atmospheric pressure. The loss of $SO_2$ over time $\Delta t$, $\Delta SO_{2g,i}$, is solved analytically (Eq. 13). $[SO_{2,t=0}]_g$ and $[X_{i,t=0}]_g$ are the mixing ratios (mol mol$^{-1}$) for $SO_{2(g)}$ and $X_{i(g)}$ at the beginning of this time step. The grid-average losses of $SO_{2(g)}$ from all seven reactions are limited by the availability of $SO_{2(g)}$ within the cloud fraction $f_c$.

$$k_{1,aq,i} = R_{aq,i}/[SO_2^T]_{aq} \text{ and } k_{2,aq,i} = R_{aq,i}/\left([SO_2^T]_{aq}[X_i]_{aq}\right) \qquad (10)$$

$$k_{1,g,i} = k_{1,aq,i}H_{SO2}^* f_{g,SO2}LRT \text{ and } k_{2,g,i} = k_{2,aq,i}H_{SO2}^* f_{g,SO2}H_{Xi}f_{g,Xi}PLRT \qquad (11)$$

$$f_{g,SO2} = \left(1 + H_{SO2}^* LRT\right)^{-1} \text{ and } f_{g,Xi} = (1 + H_{Xi}LRT)^{-1} \qquad (12)$$

$$\Delta SO_{2g,i} = \begin{cases} [SO_{2,t=0}]_g[1 - \exp(-k_{1,g,i}\Delta t)], \text{ 1}^{st}\text{ order} \\[2mm] \dfrac{[SO_{2,t=0}]_g[X_{i,t=0}]_g(C-1)}{[SO_{2,t=0}]_g C - [X_{i,t=0}]_g}, C = \exp\left[\left([SO_{2,t=0}]_g - [X_{i,t=0}]_g\right)k_{2,g,i}\Delta t\right], \text{ 2}^{nd}\text{ order} \end{cases}$$

(13)

Since multiple in-cloud reactions consume $SO_2$ simultaneously, it is important to allow them to compete effectively and fairly. As shown in Eq. 13, the contribution of the $i^{th}$ reaction to the total $SO_{2(g)}$ loss depends on its rate constant ($k_{1,g,i}$ or $k_{2,g,i}$), its relative abundance ($[X_{i,t=0}]_g/[SO_{2,t=0}]_g$), and the choice of $\Delta t$. Ideally, $\Delta t$ should be smaller than the lifetime ($\tau_i$) of $SO_{2(g)}$ for any $i^{th}$ reaction. $\tau_i$ is the inverse of the pseudo-first-order rate constant, $\tilde{k_{1,g,i}}$, which equals to $k_{1,g,i}$ for a first-order reaction and to $k_{2,g,i}[X_{i,t=0}]_g$ for a second-order reaction. Figure 3 shows the probability density distributions of the calculated $\tilde{k_{1,g,i}}$ and the

total rate constant for the seven reactions, $\sum_{i=1}^{7} \tilde{k_{1,g,i}}$, in the lower troposphere for a randomly selected week in boreal summer. $\tilde{k_{1,g,i}}$ (and thus $\tau_i$) can vary by several orders of magnitude in different model grid cells. Notably, there is a > 50% possibility that the lifetime of $SO_{2(g)}$ is smaller than 20 min, the $\Delta t$ used in this simulation. The rapid consumption of $SO_{2(g)}$ is mainly via $O_3$ and $H_2O_2$, as shown in Fig. 3 and Table S1 (statistics of probability distributions). This means that using $\Delta t = 20$ min for the sulfur chemistry will in general lead to an underestimation of the contribution of $O_3$ and $H_2O_2$ and an overestimation of the

importance of the other reactants such as HCHO. An example is provided in Text S1 to conceptually explain the effect of $\Delta t$ on the competition of different reactions. We conduct a sensitivity simulation in which $\Delta t$ is set to 10 min and, as we expect, the $SO_4^{2-}$ concentrations through the cloud $O_3$ chemistry increase significantly (Fig. S3). A simple way to solve this problem is to reduce $\Delta t$. The possibility of $\tau < \Delta t$ decreases to only 4% when $\Delta t = 1$ min (Fig. 3 and Table S1). Also, most (> 80%) of





the cases of $\tau < 1$ min arise from the rapid reaction of $SO_3^{2-}$ with $O_{3(aq)}$ when cloud water pH is high. The remaining cases are from the reactions of $SO_3^{2-}$ with $HOBr_{(aq)}$ and $HCHO_{(aq)}$. The other four reactions can hardly lead to $\tau < 1$ min. We change the time step to 1 min when calculating in-cloud $SO_2$ titration in the default simulation (Sect. 2.5).

Another issue in the control simulation is, in a partly cloudy ($0 < f_c < 1$) model grid, that the mixing of air between the cloudy fraction ($f_c$) and the cloud free fraction ($1-f_c$) occurs in the same timescale as the chemistry time step of the model (Holmes et al., 2019). In each time step, the grid-average loss of $SO_{2(g)}$ from all in-cloud reactions cannot exceed the amount of $SO_{2(g)}$ available within the cloudy fraction and at the beginning of this time step (Eq. 14). This so-called "cloud partitioning method" is unphysical as the entrainment/detrainment rates are affected by the setting of the chemistry time step (Holmes et al., 2019).

Since many chemical transport models such as GEOS-Chem do not resolve individual clouds, Holmes et al. (2019) developed a more realistic and stable "entrainment-limited uptake" method, which accounts for cloud entrainment/detrainment within the chemical rate expression. We apply this method to the default simulation (Sect. 2.5).

$$\sum_{i=1}^{7} \Delta SO_{2\,g,i} \leq f_c \left[ SO_{2,t=0} \right]_g$$

(14)

**2.5 Default simulation**

Three major changes are made in this simulation based on the control simulation, as mentioned in Sect. 2.4. The first is applying the "entrainment-limited uptake" method developed by Holmes et al. (2019) to more realistically model the entrainments and detrainments of air in cloudy grid cells. The second is reducing the time step to 1 min when calculating cloud sulfur reactions to better quantify the competition of different chemical pathways consuming $SO_2$. The third is adding the reaction of $H_2O_2$ and

$SO_2$ in aerosol water using the new kinetic data from Liu et al. (2020). Figure S4 shows the horizontal distributions of surface $SO_4^{2-}$ and $SO_2$ concentrations in the control and default simulations, and only very small differences (4% for $SO_4^{2-}$ and 1% for $SO_2$) are found for their global average values.

In the "entrainment-limited uptake method" (Holmes et al., 2019), the first-order loss rate of $SO_{2(g)}$ in a model grid cell due to

heterogeneous cloud chemistry, $k_1$ ($s^{-1}$), depends on the cloud fraction ($f_c$), the detrainment rate ($k_c$, $s^{-1}$), and the in-cloud total pseudo-first-order rate constant, $k_{1,g}^* = \sum_{i=1}^{7} k_{1,g,i}^*$ ($s^{-1}$) (Eq. 15). As shown in Holmes et al. (2019), the entrainment/detrainment ($k_c$ term) limits its reactive uptake. In a completely cloudy condition ($f_c = 1$), Eq. 15 reduces to $k_1 = \sum_{i=1}^{7} k_{1,g,i}^*$. $k_c$ is the reverse of the in-cloud residence time of air ($\tau_c$), which varies with cloud types and ranges from 15 to 120 min for stratus and cumulus clouds (Holmes et al., 2019). We use $\tau_c = 30$ min in this work since MERRA-2 does not provide this information. A sensitivity

simulation shows that assuming a $\tau_c$ of 60 min decreases the global average surface $SO_4^{2-}$ concentration by 10%. Holmes et al.



(2019) have pointed out that future studies are needed to specify the spatiotemporal variability of $\tau_c$ in the global reanalysis datasets. Within the cloudy fraction of a model grid cell, as shown in Fig. 1 and Eq. 16, the heterogeneous reaction rates are limited by a series of resistances associated with the mass transfer processes from the gas phase to the aqueous phase, including gas-phase diffusion, transfer of the reactants across the air–water interface, and aqueous-phase diffusion (Ravishankara, 1997;

Jacob, 2000). In Eq. 16, the $\alpha_{SO2}$ term represents the limitation due to mass accommodation at the air–water interface and the $D_{g,SO2}$ term represents that due to gas-phase diffusion. A dimensionless parameter $Q$, whose expression is given by Eq. 17 (0 $< Q < 1$), is used to account for aqueous-phase mass transport limitations when calculating $k_{1,aq,i}$ and $k_{2,aq,i}$ (Eq. 10).

$$\frac{1}{k_1} = \frac{1 - f_c}{f_c k_c} + \frac{1}{f_c k_{1,g}^*} = \frac{1 - f_c}{f_c k_c} + \frac{1}{f_c \sum_{i=1}^{7} k_{1,g,i}^*}$$

(15)

$$\frac{1}{k_{1,g,i}^*} = \frac{1}{\tilde{k_{1,g,i}}} + \frac{4}{A v_{SO2} \alpha_{SO2}} + \frac{r}{A D_{g,SO2}}$$

(16)

$$Q = 3 \left( \frac{\coth q}{q} - \frac{1}{q^2} \right), \quad q = r \sqrt{\frac{\sum_{i=1}^{7} \tilde{k_{1,aq,i}}}{D_{aq}}}$$

(17)

Here, $r$ is the radius of cloud droplets and is assumed to be $10^{-5}$ m. $A$ (m$^2$ m$^{-3}$ air) is the surface area density of cloud droplets and is derived using $L$ and $r$. $v_{SO2}$ (m s$^{-1}$) is the molecular mean speed of SO$_2$ (Eq. 18). $\alpha_{SO2}$ (dimensionless) is the mass accommodation coefficient of SO$_2$ (Table 3). $D_{g,SO2}$ (m$^2$ s$^{-1}$) is the gas-phase diffusion coefficient of SO$_2$ (Eq. 19). $q$ is a dimensionless parameter determined by $r$, $D_{aq}$, and $\tilde{k_{1,aq,i}}$ (the pseudo-first-order rate constant with respect to SO$_{2(aq)}^T$ for the $i^{th}$ reaction). For a first-order and second-order reaction, $\tilde{k_{1,aq,i}}$ is equal to $k_{1,aq,i}$ and $k_{2,aq,i}[X_i]_{aq}$, respectively (Eq. 10). $D_{aq}$ is

the aqueous-phase diffusion coefficient ($10^{-9}$ m$^2$ s$^{-1}$) (Song et al., 2019a). $M_{SO2}$ (g mol$^{-1}$) represents the molar mass of SO$_2$. $\rho_{n,air}$ (molecule cm$^{-3}$) is the number density of air. $\tilde{k_{1,g,i}}$ (s$^{-1}$) is the pseudo-first-order rate constant with respect to SO$_{2(g)}$ for the $i^{th}$ aqueous-phase reaction, and equals to $k_{1,g,i}$ and $k_{2,g,i}[X_i]_g$ for the first-order and second-order reactions, respectively. In addition, as illustrated in Fig. 1, the second-order reaction rate may also be limited by the mass transfer of $X_i$. Thus, the in-cloud pseudo-second-order rate constant, $k_{2,g,i}^*$, is given by Eq. 20. $v_{Xi}$, $\alpha_{Xi}$, and $D_{g,Xi}$ are the molecular mean speed, the mass

accommodation coefficient, and the gas-phase diffusion coefficient of $X_i$, respectively. $v_{Xi}$ and $D_{g,Xi}$ are calculated similarly to Eqs. 18 and 19. $\alpha_{Xi}$ can be found in Table 3.

$$v_{SO2} = \sqrt{8RT/(\pi M_{SO2})}$$

(18)



$$D_{g,SO2} = \frac{9.45 \times 10^{13} \times \sqrt{T \times (3.47 \times 10^{-2} + 1/M_{SO2})}}{\rho_{n,air}}$$

(19)

$$\frac{1}{k_{2,g,i}^*} = \frac{1}{k_{2,g,i}} + MAX\left(\frac{4[X_i]_g}{Av_{SO2}\alpha_{SO2}} + \frac{r[X_i]_g}{AD_{g,SO2}}, \frac{4[SO_2]_g}{Av_{Xi}\alpha_{Xi}} + \frac{r[SO_2]_g}{AD_{g,Xi}}\right)$$

(20)

As mentioned in Sect. 2.4, we do not change the chemistry time step of the model ($\Delta t$ = 20 min) but only the time step (to 1 min) when identifying cloud $SO_2$ reactions. For each 1-min time step, the loss of $SO_{2(g)}$ for the $i^{th}$ reaction, $\Delta SO_{2g,i}$, is solved analytically using Eq. 13, in which $k_{1,g,i}$ and $k_{2,g,i}$ are replaced by $k_{1,g,i}^*$ from Eq. 16 and $k_{2,g,i}^*$ from Eq. 20, respectively. This change reflects the mass transport limitations. The grid-average first-order loss rate of $SO_{2(g)}$, $k_1$, is calculated using Eq. 15, in which $\sum_{i=1}^{7} k_{1,g,i}^*$ is replaced by the in-cloud total pseudo-first-order rate constant estimated from $\sum_{i=1}^{7} \Delta SO_{2g,i}$. The grid-average loss of $SO_{2(g)}$ and the contributions of different reactions are calculated then using $k_1$ and $\Delta SO_{2g,i}$. The mixing ratios of the relevant chemical species are updated at the end of this 1-min time step and used as initial condition for the next. The calculations are repeated 20 times in a chemistry time step.

The cloud water pH and rate constants for the heterogeneous reaction of $SO_{2(g)}$ and $X_{i(g)}$ are calculated only at the beginning of each chemistry time step. We conduct a sensitivity simulation that redoes cloud $SO_2$ calculations using the cloud water pH and rate constants estimated at the end of each chemistry time step. The resulting change is insignificant (global mean $SO_4^{2-}$ concentration decreases by < 2%). The aqueous-phase sulfur reactions are hard-coded into the model. Ideally, further model development of cloud chemistry should apply the advanced numerical solvers generated by the Kinetic PreProcessor (KPP), which may not only allow a full coupling of gas-phase and cloud chemistry but also make it easier for the model to incorporate additional aqueous reactions (Fahey et al., 2017; Personal communication, Viral Shah, 18 December 2019).

The implementation of sulfur chemistry in aerosol water is similar to that for cloud sulfur chemistry. As shown in Fig. 1, the heterogeneous reaction rates are also controlled by the mass transfer of reactants from the gas to the aqueous phase. The difference is that aerosols (and aerosol water) can be considered evenly distributed in a model grid cell, and it is unnecessary to include the entrainment/detrainment processes. The major difficulty in parameterizing the aerosol water sulfur chemistry is the lack of suitable reaction rate constants. Liu et al. (2020) have recently found that the high ionic strength of deliquesced aerosols significantly enhances the rate constants for the reaction of $H_2O_2$ and $SO_2$. The enhancement factor (EF) relative to its rate constant in dilute solutions is derived by fitting the data in Liu et al. (2020) as a function of the molality-based ionic strength, $\mu_b$ (mol kg$^{-1}$) (Table 1). The water content, pH, $\mu_b$, and the absorbed water volume fraction of inorganic aerosols are





calculated by the ISORROPIA II model (Sect. 2.2). The aerosol water volume fraction of 0.25 is used as a threshold for the occurrence of aqueous reactions as it governs the transition of aerosols to a liquid state (Bateman et al., 2016).

**2.6 Sensitivity simulations**

In addition to the control and default simulations, we conduct ten sensitivity simulations to investigate the key factors leading

to uncertainty in the modeled HMS concentrations. As shown in Table 4, all these sensitivity simulations are based on the default simulation. HiKf, HiKd, and HiOH make changes to HMS formation, decomposition, and oxidation, respectively, in heterogeneous cloud chemistry. HiKf uses the high $k_f$ instead of the low $k_f$ in the default simulation (Sect. 2.1.1). HiKd increases $k_d$ by a factor of 2, the upper limit of its estimate (Sect. 2.1.2). HiOH increases $[OH]_{aq}$ by a factor of 10, matching its average value in current multiphase models (Sect. 2.1.3). CWpH considers less ions, i.e., $NH_4^+$, $H^+$, $OH^-$, $SO_4^{2-}$, $NO_3^-$,

$HSO_3^-$, $SO_3^{2-}$, $HCO_3^-$, $CO_3^{2-}$, HMS, and $CH_3SO_3^-$, in cloud water pH calculations.

AWOH, AWK0, and AWKE examine the potential role of aerosol water in heterogeneous HMS chemistry (Table 4). Since the rate constants of HMS chemical reactions in concentrated solutions have not been determined experimentally, we have to make assumptions about these data. AWOH implements the oxidation of HMS by OH in aerosol water and assumes the same

rate constant as those for cloud water. AWK0 adds the formation and decomposition of HMS in aerosol water also using the rate constants for cloud water. Theoretically, we anticipate that the rate constant of HMS formation, $k_f$, in concentrated solutions should be enhanced relative to dilute solutions (Song et al., 2019a), similar to the situation found for the reaction of $H_2O_2$ and $SO_2$. AWKE arbitrarily increases $k_f$ by the same EF for the $H_2O_2$ and $SO_2$ reaction (Table 1). The implementation of the above chemical reactions of HMS in aerosol water follows the approach described in Sect. 2.5.

Three sensitivity simulations, HiNH3, HiFA, and AppHet, focus on East Asia (Table 4). $SO_2$, HCHO, and $NH_3$ emissions may influence the modeled HMS. Recent studies have shown, although $SO_2$ emissions are well understood, that there may be large uncertainties in emissions of $NH_3$ and HCHO in China (Pan et al., 2018; Kong et al., 2019; Liu et al., 2019; Song et al., 2019a). An inverse study found that the MEIC inventory underestimated $NH_3$ emissions by 30% nationally and by > 40% in eastern

and central regions using observations over the same time period as our study (Kong et al., 2019). HiNH3 increases the anthropogenic emissions of $NH_3$ in MEIC by 50%. Less information is available regarding the emissions of HCHO due to its sparse observations and complex chemistry. Model–observation comparisons in Beijing suggested a strong underestimation of HCHO emissions during winter (Song et al., 2019a). Mobile and residential emission sources may be responsible for its underestimation (Jaeglé et al., 2018; Song et al., 2019a). HiFA increases HCHO emissions from the transportation and

residential sectors by a factor of 5. Chemical transport models commonly underestimate $SO_4^{2-}$ during winter haze episodes in China, and thus some studies have adopted an apparent heterogeneous parameterization for $SO_2$ reactive uptake in order to compensate for the missing $SO_4^{2-}$ (Zheng et al., 2015; Cheng et al., 2016; Wang et al., 2016; Liu et al., 2019; Li et al., 2020).





This parameterization is applied in AppHet during the cold season, in which the reactive uptake coefficient of $SO_2$ increases from $2 \times 10^{-5}$ to $5 \times 10^{-5}$ with $50\% < RH \leq 100\%$ (Zheng et al., 2015).

## 3 Results and discussion

### 3.1 Spatial and seasonal distributions in the default simulation

The horizontal distributions of HMS concentration in the surface layer and the vertical profiles of its zonal average are shown in Fig. 4 (DJF: December–January–February and JJA: June–July–August) and Fig. S5 (MAM: March–April–May and SON: September–October–November). The concentration unit is µg $sm^{-3}$, where 1 $sm^3$ equals 1 $m^3$ at standard temperature and pressure (273.15 K and 1013.25 hPa). The molar ratio of HMS to sulfate, also shown in Fig. 4 and Fig. S5, is a useful metric to assess the significance of HMS in sulfur chemistry. Higher HMS concentrations and HMS/sulfate ratios are found over the continental regions in the Northern Hemisphere. The vertical profiles indicate that most HMS exists in the lower troposphere. These features are expected because the precursors of HMS, $SO_2$ and HCHO, are more abundant in these regions compared with elsewhere (Fig. S6).

The surface HMS concentrations and HMS/sulfate molar ratios exhibit distinct seasonal patterns with maxima in DJF (boreal winter) and minima in JJA (boreal summer), and thus our analyses focus on these two seasons. It is noted that there are hotspots of HMS in JJA in Siberia that are linked to massive forest fires in that region in July 2016 (Sitnov et al., 2017). As highlighted in Fig. 4, three regions with relatively high HMS levels, East Asia (EA), Europe (EU), and North America (NA), are selected for quantitative analysis. Figure 5 shows the statistics of HMS levels from the default simulation with comparisons to other simulations. The average HMS concentrations in DJF (JJA) are 0.59 (0.09), 0.16 (0.013), 0.055 (0.015) µg $sm^{-3}$, for EA, EU, and NA, respectively. The average HMS/sulfate ratios in DJF (JJA) are 0.09 (0.01), 0.06 (0.006), 0.04 (0.009), for EA, EU, and NA, respectively. The wintertime East Asia, northern China in particular, has both the highest HMS concentration and highest HMS/sulfate ratio. The average HMS concentrations (1–3 µg $sm^{-3}$) and HMS/sulfate ratios (0.1–0.2) are found during the winter season in northern China (Fig. S7).

As mentioned in Sect. 1, previous studies have suggested that the formation and existence of HMS in the condensed phase are favored generally under the following conditions: high precursor ($SO_2$ and HCHO) concentrations, low photochemical oxidant levels, low temperature, abundant aqueous water, and moderate acidity (Munger et al., 1984, 1986; Moch et al., 2018; Song et al., 2019a; Ma et al., 2020). The seasonal variability of HMS does not follow that of the precursor levels (Fig. S6). The seasonal variation of the geometric mean of the two precursors, $\sqrt{SO_2 \times HCHO}$, is weak because of their opposite seasonality (more $SO_2$ but less HCHO in winter). The cloud liquid water content ($L$) in the lower troposphere shows a spatial distribution with higher values over the ocean and lower values over land (Fig. S8). There is no consistent seasonal pattern of $L$ between DJF and JJA over the three regions (EA, EU, and NA). The modeled cloud water pH exhibits a seasonal difference. The average pH in DJF



(JJA) is 4.3 (5.8), 4.7 (5.6), and 4.7 (5.7) for EA, EU, and NA, respectively. The higher pH in JJA is related to more abundant gaseous $NH_3$ (Fig. S8), given the buffer capacity of $NH_3$ in moderating the acidity of atmospheric condensed water (Song et al., 2019b).

One of the above-mentioned factors favoring HMS is the moderate acidity. This term is somewhat ambiguous but is used to represent the pH range that allows for relatively rapid formation and slow decomposition of HMS. We show in Fig. 2 that both $k_f$ and $k_d$ increase with pH. The lifetime of HMS with respect to decomposition is about 60, 6, and 0.6 hours at pH 5, 6, and 7, respectively, at 298 K, and is even larger at lower $T$. For the range of pH in the three regions (its average from 4.3 to 5.8), the decomposition of HMS is so slow that its chemical equilibrium is difficult to achieve. Accordingly, the modeled HMS levels are predominantly controlled by formation kinetics. This is supported by the results from the HiKd simulation, in which $k_d \times$ 2 makes little difference in the modeled HMS compared to the default simulation (Fig. 5). The higher cloud pH in JJA should lead to faster HMS formation rates than those in DJF. However, in the default simulation, the modeled HMS levels show an opposite pattern. This is believed to be linked to the different photochemical oxidizing abilities in the two seasons. Globally, the two main aqueous oxidants for $SO_2$ are $O_3$ and $H_2O_2$, which compete with HCHO. The competition of different pathways

can be influenced by the levels of these gases, $T$ (changing gas solubilities and rate constants), and pH (changing rate constants). $O_3$, $H_2O_2$, and HCHO all have higher concentrations in JJA (Fig. S6). The lower $T$ in DJF favors the $H_2O_2$ reaction most and the $O_3$ reaction least. Notably, the response of the $O_3$ reaction to pH is essentially the same as that for HCHO since both react rapidly with $SO_3^{2-}$. The HCHO + $SO_2$ reaction is significant only when the two photochemical oxidants are inefficient.

**3.2 Difference between the default and control simulations**

Compared with the default simulation, the control simulation realizes very different spatial and seasonal distributions of HMS concentrations and HMS/sulfate molar ratios (Fig. S9). Figure 6 shows the differences in surface HMS concentrations for DJF and JJA. The corresponding information for MAM and SON is presented in Fig. S10. Two features are evident. First, a weak seasonality is found for the control simulation, but for the default simulation, HMS is much more abundant in DJF. Second, the control simulation predicts significantly higher HMS concentrations almost everywhere except in parts of East Asia and

Europe in DJF. Interestingly, the only region where the default simulation gives higher HMS concentrations is wintertime in northern China, the focus of several studies (Moch et al., 2018; Song et al., 2019a; Ma et al., 2020). Specifically, as shown in Fig. 5, the average HMS concentrations modeled by the control simulation in DJF (JJA) are 0.60 (0.70), 0.22 (0.12), 0.14 (0.11) µg sm$^{-3}$, respectively, for East Asia (EA), Europe (EU), and North America (NA). The average HMS/sulfate ratios inferred from the control simulation in DJF (JJA) are 0.09 (0.14), 0.08 (0.06), 0.09 (0.07), respectively, for EA, EU, and NA.

As described in Sect. 2.5, based on the control, the default simulation improves the representations of heterogeneous cloud sulfur chemistry in the model by applying the "entrainment-limited uptake" method from Holmes et al. (2019) and by reducing the time step when calculating aqueous sulfur reactions. These changes allow for a more realistic simulation of entrainments



and detrainments of air in partly cloudy grid cells and for an effective competition of different aqueous reactions consuming $SO_2$. In the control simulation, the time step for calculating in-cloud sulfur reactions is the same as the chemistry time step of the model, $\Delta t = 20$ min. But there may be a > 50% possibility that the lifetime of in-cloud $SO_2$ is less than this $\Delta t$, as shown in Fig. 3 and described in Sect. 2.4. Given that the main reactants with in-cloud $SO_2$ are $O_3$ and $H_2O_2$, this setting leads to a

general underestimation of the contribution of $O_3$ and $H_2O_2$ and an overestimation of importance of the minor reactants such as HCHO. The bias is larger in JJA than in DJF, as suggested by the probability distribution statistics for the in-cloud lifetime of $SO_2$ (Table S1).

## 3.3 Key uncertain factors

Section 2.6 and Table 4 describe the ten sensitivity simulations we conduct with an aim to find out the key parameters and

processes leading to HMS modeling uncertainties. All of these simulations are modified based on the default simulation and can be classified into three groups: heterogeneous cloud chemistry (HiKf, HiKd, HiOH, and CWpH); heterogeneous aerosol water chemistry (AWOH, AWK0, and AWKE); and East Asia only (HiNH3, HiFA, and AppHet). A comparison of the surface HMS concentrations and HMS/sulfate molar ratios from these sensitivity simulations is provided in Fig. 5, focusing on three regions (EA, EU, and NA) and two seasons (DJF and JJA).

First, we examine the sensitivity simulations in terms of the formulation of heterogeneous cloud chemistry. HiKf, HiKd, HiOH, and CWpH make changes in HMS formation, decomposition, oxidation, and cloud water pH calculations, respectively. The surface HMS concentrations and HMS/sulfate ratios in the latter three indicate relative differences of less than ±20% compared to the default simulation. However, HiKf shows a very large increase, by a factor of 2 to 6, in modeled HMS. This is expected

since the high and low $k_f$ differ by a factor of about 3 at pH < 2 and by a factor of about 6 at pH > 4 (Sect. 2.1.1). As described in Sect. 2.1.3, the formation of HMS and its oxidation by OH represent an indirect oxidation pathway for $SO_2$. The sulfate yield, defined as the number of $SO_4^{2-}$ ions produced due to each attack of OH on HMS, is set to 2 in our simulations. The small difference between the HiOH and default simulations suggests that this indirect pathway should be insignificant.

Second, three sensitivity simulations are conducted for East Asia, as it is found most suitable for the existence of HMS. HiNH3 and HiFA increase the concentrations for modeled HMS in DJF by 60% and 20%, respectively, whereas the concentrations by AppHet are decreased by about 30%. The changes due to HiNH3 and HiFA are much smaller in JJA (Fig. 5). The increase of HMS in HiNH3 can be attributed to higher cloud water pH, and its decrease in AppHet should be related to a decrease in $SO_2$ available for cloud chemistry. Interestingly, HiNH3 increases the HMS/sulfate ratios in DJF by only 20%. The higher cloud

water pH enhances the formation of sulfate through the pH-sensitive pathways such as the reaction of $SO_2$ with $O_3$.

Third, AWOH, AWK0, and AWKE explore the potential role of heterogeneous aerosol water HMS chemistry. The challenge of modeling aerosol water HMS chemistry is the lack of its reaction rate constants in concentrated aqueous solutions. We use





the rate constants from dilute solutions in AWOH and AWK0. The oxidation of HMS by OH in aerosol water leads to losses of 10–20% (DJF) and 40–60% (JJA) (Fig. 5). The formation and decomposition of HMS in aerosol water result in negligible changes in the modeled HMS concentrations, as shown in Fig. 7 (DJF) and Fig. S11 (the other seasons). Results from AWKE suggest that aerosol water might play a role in the formation of HMS only when the $k_f$ is strongly enhanced in concentrated

solutions like the rate constant of the $SO_2$ reaction with $H_2O_2$.

Overall, our sensitivity simulations suggest that the key uncertain parameter in the model is $k_f$. Based on existing experimental results, the low value for $k_f$ is most reasonable (Sect. 2.1.1), but we cannot rule out the possibility of higher values. The key uncertain process is modeling the aerosol water chemistry of HMS in the absence of reliably defined rate constants.

**3.4 Comparison with observations**

The observations of HMS in ambient aerosols are sparse and we provide here a comparison between the available observations and two model simulations (control and default) in Table 5. Since these observations have been collected over the past three decades while our simulations cover only one year, it is more appropriate to use the molar ratios of $HMS/SO_4^{2-}$ or HMS/MSA rather than absolute HMS concentrations. Among the observations shown in Table 5, the highest $HMS/SO_4^{2-}$ ratio of 11% has

been found in winter in Beijing by Ma et al. (2020). Model results from both default and control simulations agree well with this observed ratio. Less HMS was observed in other regions, including New Mexico (USA), Germany, and Osaka Bay (Japan). The default simulation overestimates the $HMS/SO_4^{2-}$ or HMS/MSA ratios by a factor of 2–3, whereas the control simulation overestimates these ratios by an order of magnitude. Given the various sources of uncertainty in the model and the mismatches between the observations and global simulations, we conclude that the default simulation reasonably reproduces the available

HMS observations in different regions.

A more detailed comparison of the model with observations in Beijing is provided below. The observations in Ma et al. (2020) cover 73 days in winter and 11 polluted days in other seasons. The data for the other seasons is presented only in their discussion paper. Because of the coarse resolution of global model, we do not expect our simulations to capture the day-to-day variability that is observed at a single site. Accordingly, we examine the ability of our simulations to reproduce the observed relationships

between HMS and its influencing factors. Figure 8 provides scatter plots of HMS concentrations (and $HMS/SO_4^{2-}$ ratios) versus two variables ($O_3$ and RH) and compares the data from observations and model simulations (control and default). The level of $O_3$ represents photochemical oxidizing capacity and RH may indicate the abundance of aqueous water in the lower troposphere.

We find a similar relationship between HMS and $O_3$ from the observations and default simulation (Fig. 8). Significant HMS levels are observed and modeled only under low $O_3$ conditions (< 20 ppb). However, the control simulation obtains another cluster of days with high HMS levels when $O_3$ is abundant (> 40 ppb). This cluster is linked to the inappropriate representation of heterogeneous cloud sulfur chemistry in the control simulation. The large time step for $SO_2$ titration excessively favors the





reaction of SO$_2$ with HCHO, as described in Sect. 2.4 and Fig. 3. It should be noted in Ma et al. (2020) that only 11 daily samples had O$_3$ levels larger than 20 ppb. There might be a possibility that the days with high levels of both HMS and O$_3$ were missed in their sampling coverage, but we think it is unlikely given the rapid oxidation of SO$_2$ by the photochemical oxidants (gaseous OH and aqueous O$_3$ and H$_2$O$_2$) under such conditions.

The scatter plots of HMS and RH show a similar exponential-like relationship in the observations and model simulations (Fig. 8). Such an exponential-like relationship has been interpreted to support the hypothesis that HMS is produced through aerosol water (Song et al., 2019a; Ma et al., 2020). This is because the amount of aerosol water also exhibits an exponential relationship with RH (Song et al., 2018, 2019b). Interestingly, our model simulations using aqueous clouds as the only medium can obtain

a similar relationship between HMS and RH, which reduces the credibility of this aerosol water hypothesis. Global atmospheric models, including the numerical weather prediction model employed in the MERRA-2 reanalysis, are usually not capable of resolving sub-grid cloud processes, and cloud properties are parameterized using an RH-related statistical scheme (Molod, 2012; Molod et al., 2015). Thus, it is not surprising to find a relationship between RH and HMS in the simulations.

### 3.5 Knowledge gained and remaining gaps

The different spatiotemporal patterns of the HMS levels modeled by the control and default simulations indicate the importance of an appropriate representation of heterogeneous cloud sulfur chemistry. The default simulation better reproduces the limited available observations of HMS in different regions of the world. Our modeling suggests that photochemical oxidizing capacity is a key influencing factor for HMS formation because it affects the competition of HCHO with oxidants (e.g., O$_3$ and H$_2$O$_2$) for SO$_2$. This factor is partly responsible for the distinct seasonality in HMS modeled by the default simulation. On a regional

scale, the most suitable place for the formation and existence of HMS is parts of East Asia in the lower troposphere during the cold season. Aqueous clouds are the major medium for HMS chemistry since the model simulations can reasonably reproduce both the observed HMS levels and the relationship between in situ HMS and RH when assuming this as the only medium. Aerosol water may play a role if the rate constant of HMS formation is greatly enhanced in concentrated solutions. This finding is consistent with several studies (Jacob, 1986; Olson and Hoffmann, 1989; Whiteaker and Prather, 2003; Moch et al., 2018).

The observations of HMS are sparse, and more data are required to validate the model. The quantification of HMS in different seasons and over different photochemical conditions is particularly valuable.

Two knowledge gaps are identified from our sensitivity simulations. First, the key uncertain factor in the model is $k_f$, the rate constant for HMS formation. Large discrepancies exist among existing laboratory experiments (Fig. 2), and future laboratory

studies are required to narrow its uncertainty. Second, the lack of kinetic and thermodynamic data for HMS chemistry in concentrated solutions poses a key challenge to modeling HMS processing in aerosol water, and new laboratory studies are needed. Also, we did not consider the uncertainty in the meteorological reanalysis. It is unknown whether the model results are sensitive to cloud distributions and properties. Although MERRA-2 assimilates extensive observations and represents the



atmospheric states accurately, cloud properties are modeled exclusively. Studies have shown biases in seasonal and spatial variations of cloudiness when comparing the reanalysis data with lidar and satellite observations (Kennedy et al., 2011; Stengel et al., 2018; Miao et al., 2019).

Recently, the quantum chemical calculations by Chen and Zhao (2020) suggested that hydroxymethyl sulfite (HMSi), an isomer of HMS, might also be produced by an aqueous reaction of HCHO and $SO_2$. The laboratory experiments of De Haan et al. (2020) demonstrated that HMS was one of the major products from the aqueous processing of glyoxal monobisulfite ($CH(OH)_2CH(OH)SO_3^-$), the adduct of glyoxal and $SO_2$. The new mechanisms need to be considered in future model studies.

**4 Conclusion**

Based on appropriate implementation of heterogeneous HMS chemistry and assuming aqueous clouds as the only medium, the global GEOS-Chem model can reasonably reproduce the limited available observations of HMS among different regions. The modeled HMS concentrations and HMS/sulfate ratios show a clear seasonal pattern with higher values in the cold period. The spatial distributions of HMS in descending order are East Asia, Europe, and North America. Our model simulations find the

highest average HMS concentrations (1–3 µg m$^{-3}$) and HMS/sulfate molar ratios (0.1–0.2) in northern China during the winter season. Photochemical oxidizing capacity affects the competition of HCHO with oxidants (e.g., $O_3$ and $H_2O_2$) for $SO_2$, and is a key factor influencing HMS formation. Aqueous clouds act as the primary medium for HMS chemistry while aerosol liquid water could play a role if the rate constant for HMS formation is greatly enhanced.

This study identifies future research needs. Laboratory experiments should reduce the uncertainty in the formation rate constant of HMS and determine the kinetics for HMS chemistry in concentrated solutions. More field observations of HMS, especially its quantification in different seasons and photochemical conditions, are helpful to validate the model. The coarse resolution of the global model does not allow it to capture day-to-day observations at a single site, and we are preparing another paper to demonstrate the capacities of regional model with a finer resolution to reproduce individual haze events in northern China.

*Code and Data availability*. The standard GEOS-Chem model is available at: https://doi.org/10.5281/zenodo.3860693. The code changes made in this study are available at: https://github.com/shaojiesong/GC1210_sulfchem_Song2020. The laboratory and observational data used in this study are all obtained from published papers and books.

*Supplement*. The supplement related to this article is available online at:





*Author contributions*. SS initiated the study, carried out analysis, and wrote the initial draft. All authors helped interpret the data, provided feedback, and commented on the manuscript.

*Competing interests*. The authors declare that they have no conflict of interest.

*Acknowledgements*. We thank Qianjie Chen, Pingqing Fu, John Jayne, Gan Luo, J. William Munger, Chris P. Nielsen, Jingyuan Shao, Viral Shah, Xuan Wang, Yuxuan Wang, Douglas R. Worsnop, and Lin Zhang for helpful discussions.

*Financial support*. This work was supported by the Harvard Global Institute.

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



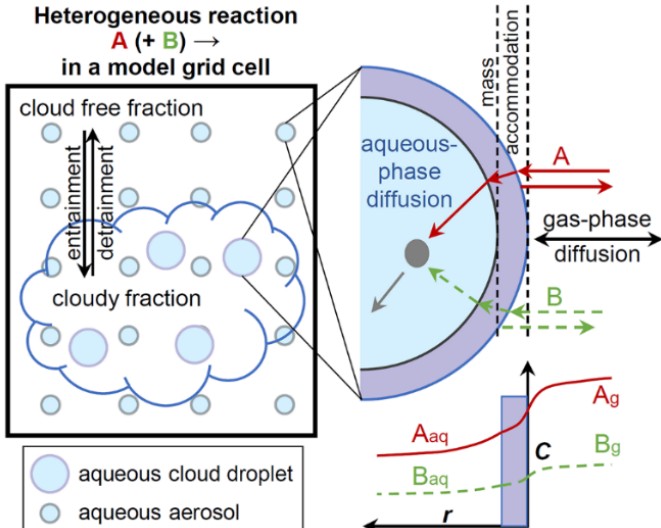

**Figure 1.** Schematic of physicochemical processes that control the heterogeneous reaction of a molecule A (with another molecule B) in a model grid cell. (**Left**). Entrainment and detrainment of air into and out from clouds. The volume occupied by aqueous clouds in the grid cell is represented by the cloud fraction ($f_c$), which is provided by the MERRA-2 meteorological reanalysis in this study. The cloud free fraction is thus $1-f_c$. Aqueous aerosols are assumed to be evenly distributed in the grid cell. For aqueous cloud droplets and aqueous aerosols, the same mass transport processes are considered and are shown in the right panel. (**Top Right**). Gas-phase, interfacial, and aqueous-phase mass transport limitations for the molecules A and B. (**Bottom Right**). Concentration ($C$) profiles of A and B are a function of radial distance ($r$) from the surface of a spherical particle. The subscripts g and aq refer to gas and aqueous phases, respectively. The concentrations are in arbitrary units and their scales are different for gas and aqueous phases. The entrainment/detrainment processes for clouds have been described in detail by Holmes et al. (2019). The right panel is adapted from Figure 4 in Ravishankara (1997).




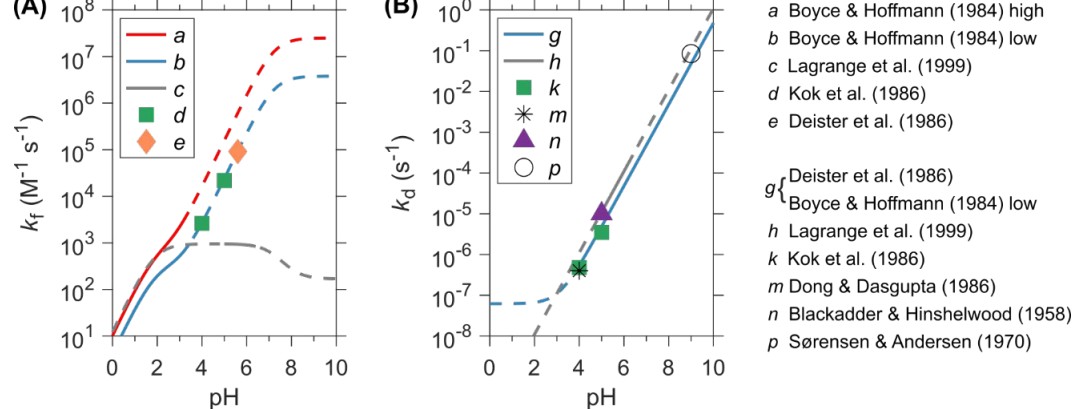

**Figure 2.** Comparison of rate constants for the formation (panel **A**, $k_f$ in $M^{-1}$ $s^{-1}$) and decomposition (panel **B**, $k_d$ in $s^{-1}$) of HMS from the available laboratory studies. Data are shown as a function of pH. Unless otherwise noted, rate constants are determined at or corrected to 25 °C and dilute condition ($\mu < 0.01$ M). For $a$, $b$, $c$, $g$, and $h$, the solid curves show the range of pH where these experiments are performed, whereas the dash curves indicate the extrapolated values. Other experiments ($d$, $e$, $k$, $m$, $n$, and $p$) are performed at discrete pH and shown by symbols. ($a$) the high $k_f$ is from Boyce and Hoffmann (1984) at $\mu = 1$ M. ($b$) the low $k_f$ is also from Boyce and Hoffmann (1984) and corrected for $\mu$ and $K_h$. ($c$) Lagrange et al. (1999): $k_f = K_h \times \left( 0.73 \times x_{HSO_3^-} + 0.13 \times x_{SO_3^{2-}} \right)$ $M^{-1}$ $s^{-1}$. ($d$) Kok et al. (1986): the reported $k_f$ is limited by the dehydration rate of $CH_2(OH)_2$, $k_{dh}$, and is thus corrected here. ($e$) is calculated using the $k_d$ and $K_{eq}$ determined by Deister et al. (1986) and is also corrected for $k_{dh}$. The calculated $k_f$ values are $2.6 \times 10^3$, $2.2 \times 10^4$, and $9.1 \times 10^4$ $M^{-1}s^{-1}$, respectively, at pH = 4, 5, and 5.6 in ($d$) and ($e$). For comparison, the extrapolation of the low $k_f$ data ($b$) are $2.7 \times 10^3$, $2.4 \times 10^4$, and $9.3 \times 10^4$ $M^{-1}s^{-1}$, respectively, at pH = 4, 5, and 5.6. ($g$) $k_d$ is calculated using the $K_{eq}$ from Deister et al. (1986) and the low $k_f$ from Boyce and Hoffmann (1984). ($h$) Lagrange et al. (1999): $k_d = 1.1 \times 10^4 \times \left( K_w / [H^+] \right)$ at $\mu = 1$ M in the presence of $H_2O_2$. ($k$) Kok et al. (1986) measured $k_d$ of $4.8 \times 10^{-7}$ and $3.5 \times 10^{-6}$ $s^{-1}$, respectively, at pH 4 and 5. ($m$) Dong and Dasgupta (1986) measured $K_{eq}$ at pH 4 and $\mu = 0.05$ M, which translated to a $k_d$ of $4 \times 10^{-7}$ $s^{-1}$. ($n$) Blackadder and Hinshelwood (1958): $k_d = 1 \times 10^{-5}$ $s^{-1}$ at pH 5 and $\mu \approx 0.1$ M. ($p$) Sørensen and Andersen (1970): $k_d = 8.5 \times 10^{-2}$ $s^{-1}$ at pH 9 and $\mu = 0.1$ M. For comparison, values of $k_d$ calculated by ($g$) are $5.4 \times 10^{-7}$, $4.9 \times 10^{-6}$, and $4.8 \times 10^{-2}$ $s^{-1}$ at pH 4, 5, and 9, respectively.

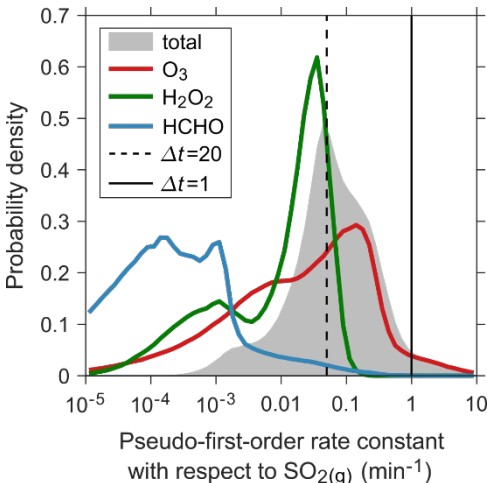

**Figure 3.** Probability density distributions of the pseudo-first-order rate constants with respect to $SO_{2(g)}$ for cloud reactions in the control simulation. The shaded area shows the sum of rate constants for the 7 reactions consuming $SO_2$. The red, green, and blue curves indicate the distributions for reactions with $O_3$, $H_2O_2$, and HCHO, respectively. Data shown are for the first week of July and in the lower troposphere (13 vertical layers above surface up to about 800 hPa). Since the chemistry time step ($\Delta t$) of this simulation is 20 min, there are 504 steps in this week. The total number of data points is $72 \times 46$ (number of $5° \times 4°$ grids) $\times 13$ (vertical layers) $\times 504 \approx 2.2 \times 10^7$. About $1.4 \times 10^7$ data points have aqueous clouds (cloud fraction $f_c > 0$), accounting for about 2/3. The probability density distributions are plotted based on these data points. The dashed and solid vertical black lines indicate the rate constants corresponding to $\Delta t$ of 20 min and 1 min, respectively.

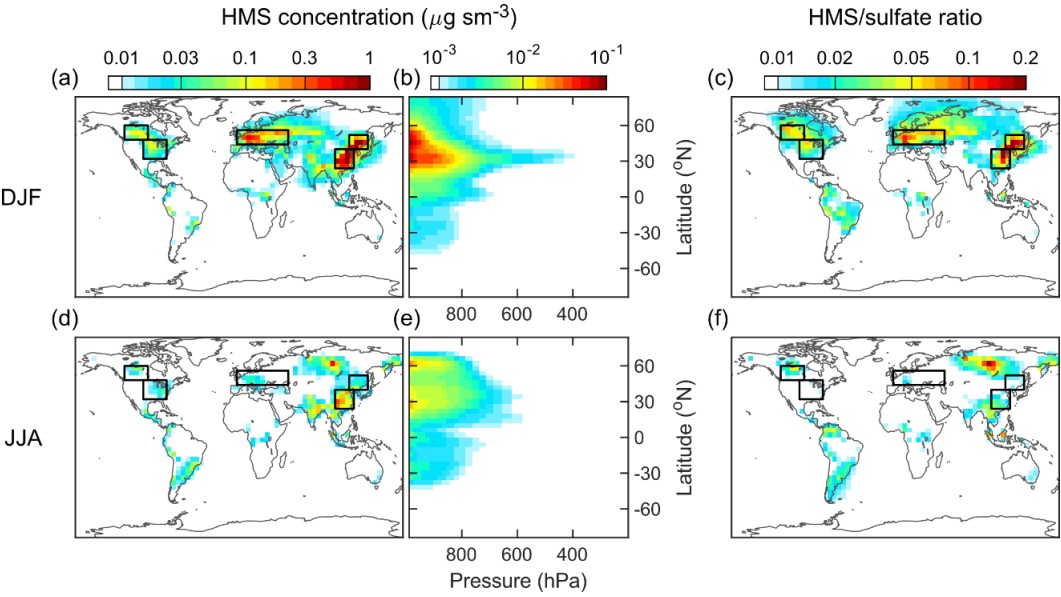

**Figure 4.** Distributions of HMS concentrations and the molar ratios of HMS to sulfate modeled by the default simulation. Top and bottom panels show results for DJF (December–January–February) and JJA (June–July–August), respectively. (a), (c), (d), and (f) are the horizontal distributions in the surface layer. (b) and (e) are the vertical distributions of the zonal averages from surface to 200 hPa. The concentration unit is µg sm$^{-3}$, where 1 sm$^3$ equals 1 m$^3$ at 273.15 K and 1013.25 hPa. The color bars are not linear and differ in the three columns. The same color bars are used for each pair of the top and bottom panels. The black-outline boxes indicate the three regions selected for quantitative analysis.



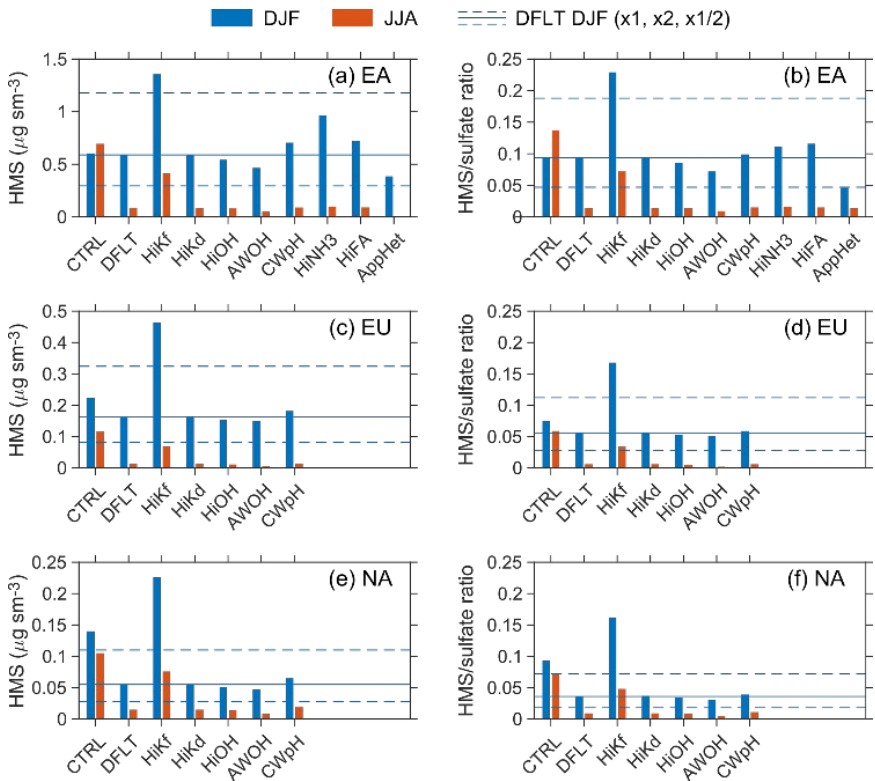

**Figure 5.** Comparison of the modeled surface HMS concentrations (left) and HMS/sulfate molar ratios (right) from different simulations for three regions and two seasons. EA, EU, and NA are East Asia, Europe, and North America, respectively. DJF and JJA represent December–January–February and June–July–August, respectively. The solid and dashed lines indicate the DJF value from the default (DFLT) simulation and its double and half. The vertical axis differs in the left panels.





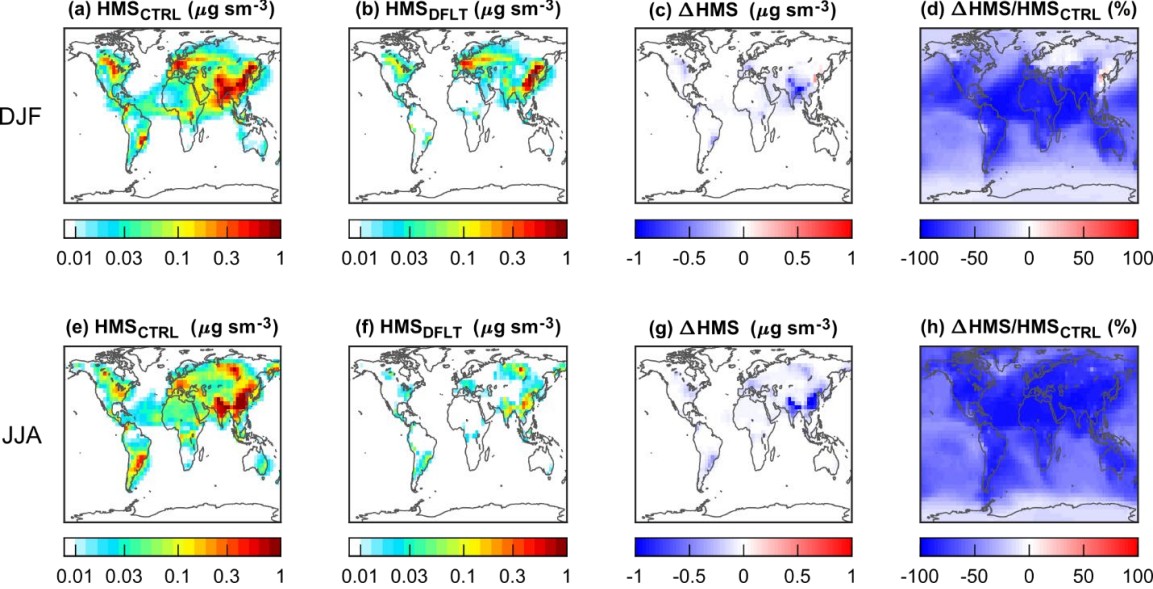

**Figure 6.** Surface concentrations of HMS from the control (CTRL) and default (DFLT) simulations in two seasons. Top and bottom panels show results for DJF (December–January–February) and JJA (June–July–August), respectively. (c) is the absolute difference between these two simulations: b−a. (d) is their relative difference: (b/a−1)×100%. Similarly, (g) is the absolute difference between (e) and (f): f−e, and (h) the relative difference between them: (f/e−1)×100%. The color bars in (a), (b), (e), and (f) are not linear.

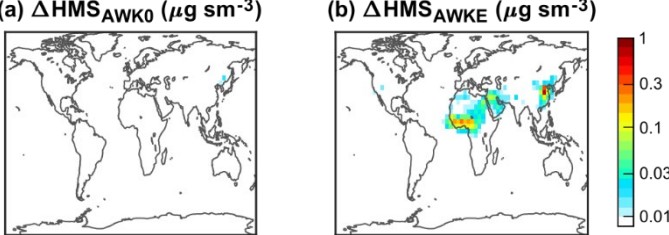

**Figure 7.** Difference in surface HMS concentrations in DJF (December–January–February) between two sensitivity simulations (AWK0 (a) and AWKE (b)) and the default simulation (DFLT). The color bar is not linear.

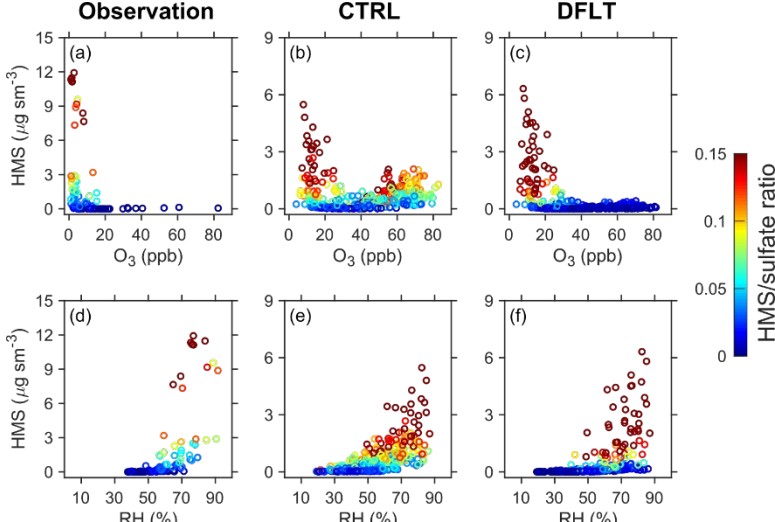

**Figure 8.** Relationship between HMS concentrations and $O_3$ mixing ratios (top) and RH (bottom) in northern China. Data from observations, the control (CTRL), and the default (DFLT) simulations are presented in (a) and (d), (b) and (e), and (c) and (f), respectively. HMS/sulfate molar ratios are indicated by the color scale. The observational data in (a) and (d) were collected in Beijing by Ma et al. (2020). There were 69 daily (and 8 half-day) samples in the 2015/16 and 2016/17 winter seasons. The model data are obtained from the grid cell covering Beijing with 366 daily samples. The vertical axes differ in the panels of observations and model simulations.


**Table 1.** Aqueous-phase reaction rate expressions.

| Reaction | Rate expression (M s$^{-1}$) | Reference and note |
|---|---|---|
| SO$_2$ + HCHO → HMS | $R_{aq} = k_f[\text{HCHO}]_{aq}[\text{SO}_2^T]_{aq} = \left(k_1 x_{\text{HSO}_3^-} + k_2 x_{\text{SO}_3^{2-}}\right)[\text{HCHO}]_{aq}[\text{SO}_2^T]_{aq}$ $k_{1,\text{high}} = 7.9 \times 10^2 \times \exp(-3000 \times (1/T - 1/298))$ M$^{-1}$ s$^{-1}$ $k_{2,\text{high}} = 2.5 \times 10^7 \times \exp(-2500 \times (1/T - 1/298))$ M$^{-1}$ s$^{-1}$ $k_{1,\text{low}} = 3.2 \times 10^2 \times \exp(-2700 \times (1/T - 1/298))$ M$^{-1}$ s$^{-1}$ $k_{2,\text{low}} = 3.8 \times 10^6 \times \exp(-2500 \times (1/T - 1/298))$ M$^{-1}$ s$^{-1}$ | Boyce and Hoffmann (1984) |
| HMS → SO$_2$ + HCHO | $k_d = 6.2 \times 10^{-8} \times \exp(-11400 \times (1/T - 1/298))$ $+ 4.8 \times 10^3 \times (K_w/[\text{H}^+]) \times \exp(-4700 \times (1/T - 1/298))$ s$^{-1}$ | Boyce and Hoffmann (1984); Deister et al. (1986) |
| HMS + OH $\xrightarrow{\text{O}_2}$ HCHO + SO$_5^-$ | $k_3 = 2.7 \times 10^8$ M$^{-1}$ s$^{-1}$ | Olson and Fessenden (1992) |
| SO$_2$ + O$_3$ → SO$_4^{2-}$ + O$_2$ | $R_{aq} = \left(k_1 x_{\text{SO}_2 \cdot \text{H}_2\text{O}} + k_2 x_{\text{HSO}_3^-} + k_3 x_{\text{SO}_3^{2-}}\right)[\text{O}_3]_{aq}[\text{SO}_2^T]_{aq}$ $k_1 = 2.4 \times 10^4$ M$^{-1}$ s$^{-1}$ $k_2 = 3.7 \times 10^5 \times \exp(-5530 \times (1/T - 1/298))$ M$^{-1}$ s$^{-1}$ $k_3 = 1.5 \times 10^9 \times \exp(-5280 \times (1/T - 1/298))$ M$^{-1}$ s$^{-1}$ | Seinfeld and Pandis (2016) |
| SO$_2$ + H$_2$O$_2$ → SO$_4^{2-}$ + H$_2$O | $R_{aq} = k_4 K_{s1} x_{\text{SO}_2 \cdot \text{H}_2\text{O}}[\text{H}_2\text{O}_2]_{aq}[\text{SO}_2^T]_{aq}$ For cloud water, $k_4 =$ $7.45 \times 10^7 \times \exp(-4430 \times (1/T - 1/298))/(1 + 13[\text{H}^+])$ M$^{-2}$ s$^{-1}$ For aerosol water, $k_4$ is multiplied by an enhancement factor EF that is dependent on $\mu_b$: $\text{EF} = \begin{cases} 1.5, 0 < \mu_b \leq 4 \\ 2.3\exp(2.4\log_{10}\mu_b - 1.2), \mu_b > 4 \end{cases}$ | Seinfeld and Pandis (2016); Liu et al. (2020); note $a$ |
| SO$_2$ + O$_2$ $\xrightarrow{\text{Mn}^{2+}, \text{Fe}^{3+}}$ SO$_4^{2-}$ | $R_{aq} = k_5[\text{H}^+]^{-0.74}[\text{Mn}^{2+}][\text{Fe}^{3+}][\text{SO}_2^T]_{aq}$ (pH < 4.2) $R_{aq} = k_6[\text{H}^+]^{0.67}[\text{Mn}^{2+}][\text{Fe}^{3+}][\text{SO}_2^T]_{aq}$ (pH ≥ 4.2) $k_5 = 3.7 \times 10^7 \times \exp(-8400 \times (1/T - 1/297)) \times 10^{-3\sqrt{\mu}}$ M$^{-2}$ s$^{-1}$ $k_6 = 2.5 \times 10^{13} \times \exp(-8400 \times (1/T - 1/297)) \times 10^{-3\sqrt{\mu}}$ M$^{-2}$ s$^{-1}$ | Shao et al. (2019); note $b$ |
| SO$_2$ + 2NO$_2$ → SO$_4^{2-}$ + 2HONO | $R_{aq} = k_7[\text{NO}_2]_{aq}[\text{SO}_2^T]_{aq}$ $k_7 = 1.4 \times 10^5$ M$^{-1}$s$^{-1}$ (pH < 5) $k_7 = 8.4 \times 10^{-3}[\text{H}^+]^{-1.444}$ M$^{-1}$s$^{-1}$ (5 ≤ pH ≤ 5.8) $k_7 = 2 \times 10^6$ M$^{-1}$s$^{-1}$ (pH > 5.8) | Cheng et al. (2016); note $c$ |
| SO$_2$ + HONO → SO$_4^{2-}$ + $\frac{1}{2}$N$_2$O | $R_{aq} = k_8[\text{H}^+]^{0.5}[\text{HNO}_2^T]_{aq}[\text{SO}_2^T]_{aq}$ $k_8 = 142$ M$^{-1}$s$^{-1}$ | Martin et al. (1981); note $d$ |
| SO$_2$ + HOBr → SO$_4^{2-}$ + HBr | $R_{aq} = k_9 x_{\text{SO}_3^{2-}}[\text{HOBr}]_{aq}[\text{SO}_2^T]_{aq}$ $k_9 = 5 \times 10^9$ M$^{-1}$ s$^{-1}$ | Troy and Margerum (1991); note $e$ |

The chemical reaction equations are used to indicate major reactants and products and may not be balanced in terms of stoichiometry and charge.

$x_{\text{SO}_2 \cdot \text{H}_2\text{O}} = [\text{SO}_2 \cdot \text{H}_2\text{O}]/[\text{SO}_2^T]_{aq} = [\text{H}^+]^2/([\text{H}^+]^2 + K_{s1}[\text{H}^+] + K_{s1}K_{s2})$

$^a$EF is obtained by fitting the experimental data shown in Fig. 2C in Liu et al. (2020). $\mu_b$ is the molality-based ionic strength (mol kg$^{-1}$).

$^b$The relationship between $k$ and $\mu$ is: $k/k^{\mu=0} = 10^{b(\sqrt{\mu}/(1+\sqrt{\mu}))} \approx 10^{b\sqrt{\mu}}$, in which $b$ is in range of −4 to −2 (Shao et al., 2019).

$^c k_7$ is believed to be the lower limit (Cheng et al., 2016).

$^d[\text{HNO}_2^T]_{aq}$ is the total dissolved HONO and NO$_2^-$.

$^e k_9$ is determined at 25 °C, $\mu = 0.5$ M. We consider the reaction of HOBr and SO$_3^{2-}$ but not the one between HOBr and HSO$_3^-$, which is included in the standard GEOS-Chem model. The original lab experiments (Liu, 2002) seemed to be interfered by Br$_2$, a stronger oxidizing reagent which also reacts with HSO$_3^-$. A recent study by Liu and Abbatt (2020) suggested that the rate constant of HOBr and HSO$_3^-$ was much lower that of HOBr and SO$_3^{2-}$.



**Table 2.** Equilibrium reactions.

| Reaction | Constant expression | Reference and note |
|---|---|---|
| $SO_{2(g)} + H_2O \leftrightarrow SO_2 \cdot H_2O$ | $H = 1.3 \times \exp(3100 \times (1/T - 1/298))$ M atm$^{-1}$ | Sander (2015) |
| $SO_2 \cdot H_2O \leftrightarrow HSO_3^- + H^+$ | $K_{s1} = 1.3 \times 10^{-2} \times \exp(2000 \times (1/T - 1/298))$ M | Seinfeld and Pandis (2016) |
| $HSO_3^- \leftrightarrow SO_3^{2-} + H^+$ | $K_{s2} = 6.6 \times 10^{-8} \times \exp(1500 \times (1/T - 1/298))$ M | Seinfeld and Pandis (2016) |
| $H_2O \leftrightarrow OH^- + H^+$ | $K_w = 1.0 \times 10^{-14} \times \exp(-6710 \times (1/T - 1/298))$ M$^2$ | Seinfeld and Pandis (2016) |
| $HCHO_{(g)} \leftrightarrow HCHO_{(aq)}$ | $H = 2.5 \times \exp(3300 \times (1/T - 1/298))$ M atm$^{-1}$ | Song et al. (2019a) |
| $HCHO_{(aq)} + H_2O \leftrightarrow CH_2(OH)_2$ | $K_h = 1.3 \times 10^3 \times \exp(3800 \times (1/T - 1/298))$ | Song et al. (2019a) |
| | $k_h = 2 \times 10^5 \times \exp(-2900/T)$ s$^{-1}$ | |
| | $k_{dh} = k_h/K_h$ s$^{-1}$ | |
| $O_{3(g)} \leftrightarrow O_{3(aq)}$ | $H = 1.13 \times 10^{-2} \times \exp(2500 \times (1/T - 1/298))$ M atm$^{-1}$ | Sander (2015) |
| $H_2O_{2(g)} \leftrightarrow H_2O_{2(aq)}$ | $H = 9.1 \times 10^4 \times \exp(6900 \times (1/T - 1/298))$ M atm$^{-1}$ | Sander (2015) |
| $H_2O_{2(aq)} \leftrightarrow H^+ + HO_2^-$ | $K = 2.2 \times 10^{-12} \times \exp(-3730 \times (1/T - 1/298))$ M | Seinfeld and Pandis (2016) |
| $NO_{2(g)} \leftrightarrow NO_{2(aq)}$ | $H = 1.3 \times 10^{-2} \times \exp(2500 \times (1/T - 1/298))$ M atm$^{-1}$ | Sander (2015) |
| $HONO_{(g)} \leftrightarrow HONO_{(aq)}$ | $H = 48 \times \exp(4800 \times (1/T - 1/298))$ M atm$^{-1}$ | Sander (2015) |
| $HONO_{(aq)} \leftrightarrow H^+ + NO_2^-$ | $K = 5 \times 10^{-4} \times \exp(-1300 \times (1/T - 1/298))$ M | Seinfeld and Pandis (2016) |
| $HOBr_{(g)} \leftrightarrow HOBr_{(aq)}$ | $H = 1.3 \times 10^2$ M atm$^{-1}$ | Sander (2015); note $a$ |
| $OH_{(g)} \leftrightarrow OH_{(aq)}$ | $H = 32 \times \exp(3700 \times (1/T - 1/298))$ M atm$^{-1}$ | Sander (2015) |

$^a$The Henry's law constant of HOBr is very uncertain, ranging from 90 to 6000 M atm$^{-1}$. HOBr can undergo acid dissociation and has a p$K_a$ of 8.65 at 25 °C. We do not consider its acid dissociation because it is only partially dissociated in the interested pH range and because of the high uncertainty of its intrinsic Henry's law constant.

**Table 3.** Mass accommodation coefficients on aqueous surfaces.

| Species | α (dimensionless) | Reference and note |
|---|---|---|
| $SO_2$ | $[1 + \exp(14.7 - 3825/T)]^{-1}$ | Boniface et al. (2000) |
| $O_3$ | 0.04 | Müller and Heal (2002); note $a$ |
| $H_2O_2$ | 0.23 | Seinfeld and Pandis (2016) |
| HCHO | 0.04 | Davidovits et al. (2006) |
| $NO_2$ | $2 \times 10^{-4}$ | Shao et al. (2019) |
| HONO | 0.09 | Davidovits et al. (2006) |
| HOBr | 0.6 | Shao et al. (2019) |

$^a$The α of $O_3$ is very uncertain with the upper limit approaches unity.



**Table 4.** Description of model simulations.

| Abbreviation | Description |
| --- | --- |
| GC12.7.0 | Standard GEOS-Chem version 12.7.0 |
| GC12.1.0 | Standard GEOS-Chem version 12.1.0 |
| CTRL | Control simulation; major changes to GC12.1.0: adding cloud HMS chemistry and cloud reactions of $SO_2$ with HONO and $NO_2$ and applying some wet process updates |
| DFLT | Default simulation; major changes to CTRL: improving treatments of entrainment/detrainment and heterogeneous cloud sulfur chemistry and adding aerosol water reaction of $SO_2$ with $H_2O_2$; shorter time step for calculating cloud sulfur reactions |
| All the ten sensitivity simulations are based on DFLT with changes in individual parameters or processes | |
| HiKf | High $k_f$ (HMS formation rate constant); the low $k_f$ is used in DFLT |
| HiKd | High $k_d$ (HMS decomposition rate constant); $k_d$ is increased by a factor of 2 |
| HiOH | High $[OH]_{aq}$ in cloud water; $[OH]_{aq}$ is increased by a factor of 10 leading to a faster HMS oxidation in clouds |
| CWpH | Cloud water pH; its calculations do not consider $Ca^{2+}$, $Mg^{2+}$, $Na^+$, $Cl^-$, $NO_2^-$, $HCOO^-$, and $CH_3COO^-$ |
| AWOH | Aerosol water HMS oxidation by $[OH]_{aq}$; the same $[OH]_{aq}$ and oxidation rate constant are used with cloud HMS chemistry |
| AWK0 | Aerosol water HMS formation and decomposition; the same $k_f$ and $k_d$ are used with cloud HMS chemistry |
| AWKE | Aerosol water HMS formation and decomposition; the same $k_d$ with cloud HMS chemistry is used whereas the $k_f$ is enhanced relative to dilute solutions by the same EF for the reaction of $SO_2$ and $H_2O_2$ in aerosol water |
| Three sensitivity simulations below focus on the region of East Asia | |
| HiNH3 | High $NH_3$ emissions; anthropogenic $NH_3$ emissions in the MEIC inventory are increased by 50% |
| HiFA | High HCHO emissions; transportation and residential HCHO emissions in the MEIC inventory are increased by a factor of 5 |
| AppHet | Apparent heterogeneous chemistry for $SO_4^{2-}$ production; it is applied over East Asia (97.5°E–152.5°E, 16°N–56°N) during the cold season (November–March) |

**Table 5.** Comparison between observations and model simulations.

| Location | Year and season | Observed molar ratio | Default | Control | Reference and note |
| --- | --- | --- | --- | --- | --- |
| New Mexico, USA | 1997 Summer | $HMS/SO_4^{2-} < 0.2\%$ | 0.3% | 2% | Dixon and Aasen (1999); note $a$ |
| Germany | 2009/10 All Year | $HMS/SO_4^{2-} = 2\%$ | 5% | 16% | Scheinhardt et al. (2014); note $b$ |
| Beijing, China | 2015/16 and 2016/17 Winter | $HMS/SO_4^{2-} = 11\%$ | 13% | 10% | Ma et al. (2020); note $c$ |
| Osaka Bay, Japan | 1998/99 Spring, Summer | $HMS/MSA = 1$ | 3 | 20 | Suzuki et al. (2001); note $d$ |

5   [a]$PM_1$ (particles smaller than 1 μm) samples of 11 were collected with each sampled for several days.
[b]Size-segregated (5 stages under 10 μm) daily aerosol samples of 154 data sets were collected at 9 sites.
[c]$PM_{2.5}$ samples of 77 were collected with 69 daily and 8 half-day samples.
[d]Size-segregated (4 stages under 7 μm) aerosol samples of 4 data sets were collected with each sampled for 11–25 days.

