# Peer review of "Global modeling of heterogeneous hydroxymethanesulfonate chemistry"

_Atmospheric Chemistry and Physics, 2020_

## Referee Comment (RC1) · Anonymous Referee #1 · 14 Aug 2020

Hydroxymethanesulfonate has been recently identified as an important organic sulfate component in atmospheric aerosols in field measurements. Current understanding for this organosulfate chemistry is largely lacking due to sparse field measurement data and modeling studies. This manuscript presents results from a modeling study of aqueous hydroxymethanesulfonate chemistry on a global scale using latest version GEOS-Chem model. Control, default, and ten specific simulations were performed under different modeling settings by incorporating comprehensive SO2 chemistry into GEOS-Chem. The results show both spatial and seasonal variations for the mass concentration and hydroxymethanesulfonate/sulfate ratio distribution and several hotspots were identified including East Asia, Europe, and North America. The paper will improve understanding of atmospheric organosulfate chemistry in aerosol composition,

in particular, for regions with high concentrations of hydroxymethanesulfonate such as northern China. The manuscript is well written and it is recommended to be published after a minor revision. Several minor comments include:

1) The differences between control and default simulations were not explicitly shown, specifically, for reaction of SO2 with H2O2, what exactly were the differences between the two simulations since in both cases k4 was taken from Liu et al. (2020) according to Table 1?

2) In control simulation, seven pathways were considered for the heterogeneous SO2 reactions. In Fig. 3, only three reactions were shown. The authors should give a clear explanation why those reactions were far more important than others. Several sentences will be beneficial to the readers in order to better understand the order of the importance for the seven reactions.

3) Table 5 shows the comparison between simulated and measured values for the HMS to sulfate or MSA. Since the techniques used for measurements of HMS were significantly different which would affect the accuracy. It would be beneficial to point this out in Sect. 3.4.

4) The sensitivity simulation was conducted using 10 min as delta t (L26 on p11). Why 10 min rather than for example 5 min was used? Using a delta t of 5 min may correspond to median probability density in a residence time range of 1 to 20 min as shown in Fig. 3.

5) Several typos or corrections: L17 on p6, not quite understand what does this mean?" under the dark conditions of sample storage and treatment"; L1 on p10, it is suggested to rewrite "the reactants' Henry's law constants" to "the Henry's law constants of the reactants"; L12 on p14, for the next? Is something missed after "next".

---

## Author Comment (AC1) · 17 Aug 2020

Dear referee #1, You comments have been addressed and please find our responses in the attached document. Feel free to let us know if you have any further comments. Thanks! Best, Shaojie

Please also note the supplement to this comment:
https://acp.copernicus.org/preprints/acp-2020-643/acp-2020-643-AC1-supplement.pdf

---

## Referee Comment (RC2) · Anonymous Referee #3 · 10 Oct 2020

This manuscript presents a global modeling study of heterogeneous chemistry of hydroxymethanesulfonate (HMS) using the GEOS-Chem model. Recently HMS has been detected in a few measurement studies and its concentrations could reach several micron grams per cubic meters. The current study incorporates HMS chemistry into a three-dimensional chemical transport model and shows the spatial and seasonal variations of the modeled HMS. It discusses the major factors, including emission, decomposition, oxidation and pH calculation, leading to the modeled patterns of HMS, which can be referred to the parameterization of other liquid phase reactions. Overall, this manuscript is very well written and should be published in this journal after addressing a few minor revisions as listed below. 1. There is a measurement paper by Wei et al. recently published (doi:10.1021/acs.estlett.0c00528). They measured HMS in

Beijing using two analytical methods (ion chromatography and UHPLC-LTQ-Orbitrap mass spectrometry). This paper should be mentioned in the introduction and comparison should also be made with the model results. 2. The authors reported the simulated pH in cloud water. What is the simulated pH in aerosol water? 3. P2L6: What is the major source of dimethyl sulfide DMS? 4. P2L10: Can the authors comment on the relative importance of other hydroxyalkylsulfonate compounds? 5. P9 R27, R28: There is a recent paper by Wang et al. (www.nature.com/articles/s41467-020-16683-x) discussing the fast oxidation of SO2 by NO2 and HONO. Are the reaction rate constants comparable to this study?

---

## Author Response (AR1)

**Authors' Response for acp-2020-643**

**"Global modeling of heterogeneous hydroxymethanesulfonate chemistry"**

**Song et al. 2020/10/25**

The authors highly appreciate the constructive comments from two anonymous referees. We have addressed these comments from anonymous referees #1 and #3 and made corresponding changes to the manuscript. We have also made additional changes reflecting the recent publications by Moch et al. (2020) doi: 10.1029/2020jd032706 and Wei et al. (2020) doi:10.1021/acs.estlett.0c00528. In this pdf document, we include a point-by-point response to the reviews, a list of all relevant changes made in the manuscript, and a marked-up manuscript version.

Comments are in black and responses are in blue.

**Response to Anonymous Referee #1**

Hydroxymethanesulfonate has been recently identified as an important organic sulfate component in atmospheric aerosols in field measurements. Current understanding for this organosulfate chemistry is largely lacking due to sparse field measurement data and modeling studies. This manuscript presents results from a modeling study of aqueous hydroxymethanesulfonate chemistry on a global scale using latest version GEOS-Chem model. Control, default, and ten specific simulations were performed under different modeling settings by incorporating comprehensive SO2 chemistry into GEOS-Chem. The results show both spatial and seasonal variations for the mass concentration and hydroxymethanesulfonate/sulfate ratio distribution and several hotspots were identified including East Asia, Europe, and North America. The paper will improve understanding of atmospheric organosulfate chemistry in aerosol composition, in particular, for regions with high concentrations of hydroxymethanesulfonate such as northern China. The manuscript is well written and it is recommended to be published after a minor revision.

We thank the referee #1 for the positive evaluation and very helpful comments. Our responses to the specific comments are provided below.

Several minor comments include:

1) The differences between control and default simulations were not explicitly shown, specifically, for reaction of SO2 with H2O2, what exactly were the differences between the two simulations since in both cases k4 was taken from Liu et al. (2020) according to Table 1?

As shown in Table 4, there are three major changes in the default simulation compared to the control simulation:
(1) improving treatments of entrainment/detrainment and heterogeneous cloud sulfur chemistry;
(2) adding aerosol water reaction of $SO_2$ with $H_2O_2$;
(3) shorter time step for calculating cloud sulfur reactions.

Regarding the $2^{nd}$ change, the control simulation includes the reaction of $SO_2$ and $H_2O_2$ in cloud droplets. This is a classic reaction pathway, and the reaction rates have been well studied and we use the rates from Seinfeld and Pandis (2016). Liu et al. (2020) recently reported the rate constants of $SO_2$ and $H_2O_2$ in concentrated aqueous solutions (which are suitable for the application in aerosol water). Thus, for aerosol water, $k_4$ is multiplied by an enhancement factor EF that is dependent on $\mu_b$ (ionic strength) (as shown in Table 1).

We have made this clearer in Sect. 2.5 when introducing the differences between the default and control simulations:

*"The third is adding the reaction of $H_2O_2$ and $SO_2$ in aerosol water using the kinetic data reported recently by Liu et al. (2020). The control simulation only includes the reaction of $H_2O_2$ and $SO_2$ in cloud water."*

2) In control simulation, seven pathways were considered for the heterogeneous SO2 reactions. In Fig. 3, only three reactions were shown. The authors should give a clear explanation why those reactions were far more important than

others. Several sentences will be beneficial to the readers in order to better understand the order of the importance for the seven reactions.

In Fig.3, we show the probability density distributions of the reaction pathways involving $O_3$, $H_2O_2$, and HCHO, as well as the sum of the seven reactions (denoted as "total"). On a global scale, the two key pathways are $O_3$ and $H_2O_2$. The other five pathways, including $TMI/O_2$, HONO, $NO_2$, HCHO, and HOBr, are minor pathways. The reason for which we show the data of the pathway involving HCHO is that this paper focuses on the modeling of HMS. In the revised manuscript, we add the data for the other 4 pathways in Fig. 3. It shows that the other 4 pathways are minor compared to those involving $O_3$ and $H_2O_2$. We have made corresponding changes in the caption of Fig. 3.

We have also added additional statements in Sect. 2.4 to demonstrate the relative importance of these seven reaction pathways:

*"The rapid consumption of $SO_{2(g)}$ is mainly via $O_3$ and $H_2O_2$, as shown in Fig. 3 and Table S1 (statistics of probability distributions). The other five reactions consuming $SO_2$ (HCHO, $TMI+O_2$, $NO_2$, HONO, and HOBr) can be considered as minor pathways."*

We hope that these changes in the revised manuscript can make the comparison of different reaction pathways clearer.

[Figure]

3) Table 5 shows the comparison between simulated and measured values for the HMS to sulfate or MSA. Since the techniques used for measurements of HMS were significantly different which would affect the accuracy. It would be beneficial to point this out in Sect. 3.4.

We have added one sentence in Sect. 3.4 when describing the measurements of HMS:

*"The measurement techniques of HMS in different studies have been different (Table 5) and their effects on the reported results are unclear. Moreover, observations of HMS may be subject to measurement artifacts that may make quantitative comparisons between model results and observations difficult (Ma et al., 2020; Moch et al., 2020)."*

We have also updated Table 5 in order to present the exact methods of HMS measurements in different studies. Please find them in the notes of Table 5.

**Table 5.** Comparison between observations and model simulations.

| Location | Year and season | Observed molar ratio | Default | Control | Reference and note |
|---|---|---|---|---|---|
| New Mexico, USA | 1997 Summer | $HMS/SO_4^{2-} < 0.2\%$ | 0.3% | 2% | Dixon and Aasen (1999); note *a* |
| Germany | 2009/10 All Year | $HMS/SO_4^{2-} = 2\%$ | 5% | 16% | Scheinhardt et al. (2014); note *b* |
| Beijing, China | 2015/16 and 2016/17 Winter | $HMS/SO_4^{2-} = 11\%$ | 13% | 10% | Ma et al. (2020); note *c* |
| Osaka Bay, Japan | 1998/99 Spring, Summer | HMS/MSA = 1 | 3 | 20 | Suzuki et al. (2001); note *d* |

[a]$PM_1$ (particles smaller than 1 µm) samples of 11 were collected with each sampled for several days. HMS detection method: ion chromatography.
[b]Size-segregated (5 stages under 10 µm) daily aerosol samples of 154 data sets were collected at 9 sites. HMS detection method: capillary electrophoresis.
[c]$PM_{2.5}$ samples of 77 were collected with 69 daily and 8 half-day samples. HMS detection method: ion chromatography.
[d]Size-segregated (4 stages under 7 µm) aerosol samples of 4 data sets were collected with each sampled for 11–25 days. HMS detection method: proton nuclear magnetic resonance ($^1$H NMR).

4) The sensitivity simulation was conducted using 10 min as delta t (L26 on p11). Why 10 min rather than for example 5 min was used? Using a delta t of 5 min may correspond to median probability density in a residence time range of 1 to 20 min as shown in Fig. 3.

As described in Section 2.2, in the standard model, the time step (also known as the transport operator duration) for species advection, vertical mixing, and convection is set to 10 min. The time step (also known as the chemistry operator duration) is 20 min for emissions, dry deposition, photolysis, and chemistry, as recommended by Philip et al. (2016). The chemistry operator duration must be an integer multiple of the transport operator duration. Consequently, we can only use 10 min in this sensitivity test.

*Philip, S., Martin, R. V., and Keller, C. A.: Sensitivity of chemistry-transport model simulations to the duration of chemical and transport operators: a case study with GEOS-Chem v10-01, Geosci. Model Dev., 9, 1683-1695, 10.5194/gmd-9-1683-2016, 2016.*

5) Several typos or corrections:

L17 on p6, not quite understand what does this mean?" under the dark conditions of sample storage and treatment";

This statement is from the review article by Tilgner and Herrmann (2018). In the bulk measurements, aerosol samples are usually collected on a filter and subsequently diluted for analysis, while cloud water is sampled in the bulk. Both samples are subject to storage for a certain time duration before analysis. The sample storage and treatment are usually in dark conditions because the we do not want to trigger new photochemical reactions during these processes. The sample storage and treatment under the dark conditions may lead to an underestimation in aqueous OH concentrations. This is because in the real atmosphere, there are replenishments of OH and other oxidants from the gas phase, whereas in the dark environment, such replenishments do not exist.

In the revised manuscript, we cite this reference by Tilgner and Herrmann at the end of this sentence. This thorough review article has also been cited several times elsewhere in this manuscript.

*Tilgner, A., and Herrmann, H.: Tropospheric Aqueous-Phase OH Oxidation Chemistry: Current Understanding, Uptake of Highly Oxidized Organics and Its Effects, in: Multiphase Environmental Chemistry in the Atmosphere, ACS Symposium Series, 1299, American Chemical Society, 49-85, 2018.*

L1 on p10, it is suggested to rewrite "the reactants' Henry's law constants" to "the Henry's law constants of the reactants";

Corrected according to this comment.

L12 on p14, for the next? Is something missed after "next".

Corrected. Added "time step" after "next".

**Response to Anonymous Referee #3**

This manuscript presents a global modeling study of heterogeneous chemistry of hydroxymethanesulfonate (HMS) using the GEOS-Chem model. Recently HMS has been detected in a few measurement studies and its concentrations could reach several micron grams per cubic meters. The current study incorporates HMS chemistry into a three-dimensional chemical transport model and shows the spatial and seasonal variations of the modeled HMS. It discusses the major factors, including emission, decomposition, oxidation and pH calculation, leading to the modeled patterns of HMS, which can be referred to the parameterization of other liquid phase reactions. Overall, this manuscript is very well written and should be published in this journal after addressing a few minor revisions as listed below.

*We thank the referee #3 for the positive evaluation and very helpful comments. Our responses to the specific comments are provided below.*

1. There is a measurement paper by Wei et al. recently published (doi:10.1021/acs.estlett.0c00528). They measured HMS in Beijing using two analytical methods (ion chromatography and UHPLC-LTQ-Orbitrap mass spectrometry). This paper should be mentioned in the introduction and comparison should also be made with the model results.

*We have mentioned this paper in the Introduction: "high mass concentrations of hydroxymethanesulfonate (HMS), a hydroxyalkylsulfonate species, have been detected in winter Beijing, China using an aerosol mass spectrometer by Song et al. (2019a), and using an improved ion chromatography method by Ma et al. (2020), and using a UHPLC-LTQ-Orbitrap mass spectrometry by Wei et al. (2020)." In fact, the measurement periods in Wei et al. (2020) and Ma et al. (2020) are overlapped and their results are also consistent. Therefore, the same comparison results are expected between the model and these two measurement studies.*

2. The authors reported the simulated pH in cloud water. What is the simulated pH in aerosol water?

*The GEOS-Chem modeling analysis uses the ISORROPIA-II thermodynamic equilibrium module to estimate the aerosol pH values. The current analysis has found that HMS is mainly formed and processed in cloud water. The sensitivity simulation that adds the relevant processed in aerosol water does not seem to change the global results significantly. For the sake of simplicity, we do not show the modeled values and distributions of aerosol water pH in this current paper.*

3. P2L6: What is the major source of dimethyl sulfide DMS?

*We have provided the major source of DMS in this sentence: "MSA is produced primarily by the oxidation of biogenic dimethyl sulfide (DMS, mainly from marine phytoplankton) and is likely the major organosulfur species in many regions over the oceans".*

4. P2L10: Can the authors comment on the relative importance of other hydroxyalkylsulfonate compounds?

*The current understanding is that HMS is the most abundant hydroxyalkylsulfonate compounds in the atmosphere. We have made this statement clear in the revised manuscript: "hydroxymethanesulfonate (HMS), the most abundant hydroxyalkylsulfonate species commonly found in the atmosphere, have been detected in winter Beijing, China". More discussions can be found in the paper Song et al. (2019a) cited in the manuscript. The relative abundance of different hydroxyalkylsulfonate HAS species depends on several factors including the level of carbonyl precursors, aqueous pH values, etc.*

5. P9 R27, R28: There is a recent paper by Wang et al. (www.nature.com/articles/s41467-020-16683-x) discussing the fast oxidation of SO2 by NO2 and HONO. Are the reaction rate constants comparable to this study?

*Thanks for the question. As shown in Table 1 of the manuscript, we use the same reaction rate constants between $SO_2$ and $NO_2$ and HONO with those used in Wang et al. (2020). In fact, both are obtained from the laboratory studies conducted in 1980s. We also cite this paper by Wang et al. (2020) in the revised manuscript.*

**Global modeling of heterogeneous hydroxymethanesulfonate chemistry**

Shaojie Song[1], Tao Ma[2], Yuzhong Zhang[3,4], Lu Shen[1], Pengfei Liu[5], Ke Li[1], Shixian Zhai[1], Haotian Zheng[1,2], Meng Gao[6], Jonathan M. Moch[1], Fengkui Duan[2], Kebin He[2], Michael B. McElroy[1]

[revised manuscript text omitted]

$$CH_2(OH)SO_3H \leftrightarrow CH_2(OH)SO_3^- + H^+ \qquad (R1)$$

$$CH_2(OH)SO_3^- \leftrightarrow CH_2(O^-)SO_3^- + H^+ \qquad (R2)$$

$$SO_2 \cdot H_2O \leftrightarrow HSO_3^- + H^+ \qquad (R3)$$

$$HSO_3^- \leftrightarrow SO_3^{2-} + H^+ \qquad (R4)$$

$$HCHO_{(aq)} + HSO_3^- \overset{k_1}{\leftrightarrow} CH_2(OH)SO_3^- \qquad (R5)$$

$$\text{HCHO}_{(aq)} + \text{SO}_3^{2-} \overset{k_2}{\leftrightarrow} \text{CH}_2(\text{O}^-)\text{SO}_3^- \tag{R6}$$

$$\text{HCHO}_{(aq)} + \text{H}_2\text{O} \leftrightarrow \text{CH}_2(\text{OH})_2 \tag{R7}$$

$$\text{HCHO}_{(aq)} + \text{SO}_{2(aq)}^T \leftrightarrow \text{HMS} \tag{R8}$$

$$K_h = [\text{CH}_2(\text{OH})_2]/[\text{HCHO}]_{aq} \tag{1}$$

$$\left[\text{SO}_2^T\right]_{aq} = [\text{SO}_2 \cdot \text{H}_2\text{O}] + [\text{HSO}_3^-] + \left[\text{SO}_3^{2-}\right] \tag{2}$$

$$K_{eq} = [\text{HMS}]/\left([\text{HCHO}]_{aq}\left[\text{SO}_2^T\right]_{aq}\right) = k_f/k_d \tag{3}$$

$$k_f = k_{10} x_{\text{HSO}_3^-} + k_{11} x_{\text{SO}_3^{2-}} \tag{4}$$

$$x_{\text{HSO}_3^-} = [\text{HSO}_3^-]/\left[\text{SO}_2^T\right]_{aq} = K_{s1}[\text{H}^+]/([\text{H}^+]^2 + K_{s1}[\text{H}^+] + K_{s1}K_{s2}) \tag{5}$$

$$x_{\text{SO}_3^{2-}} = \left[\text{SO}_3^{2-}\right]/\left[\text{SO}_2^T\right]_{aq} = K_{s1}K_{s2}/([\text{H}^+]^2 + K_{s1}[\text{H}^+] + K_{s1}K_{s2}) \tag{6}$$

**2.1.1 HMS formation**

[revised manuscript text omitted]

$$[\text{OH}]_{\text{aq}} = [\text{OH}]_{\text{g}} \times H_{\text{OH}}^* \tag{7}$$

The products of (R9) are $\text{HCHO}_{(\text{aq})}$ and $\text{SO}_5^-$. Interestingly, the net effect of HMS formation (R8) and its subsequent oxidation (R9) is the oxidation of $\text{SO}_{2(\text{aq})}^T$ by $\text{OH}_{(\text{aq})}$, which represents thus an indirect oxidation pathway for $\text{SO}_2$. The sinks for $\text{SO}_5^-$ are mainly the reactions with $\text{O}_2^-$, $\text{HCOO}^-$, and itself (R10–R12). The reaction of $\text{SO}_5^-$ and $\text{HSO}_3^-$ is slow (Jacob et al., 1989). The peroxymonosulfate radical ($\text{HSO}_5^-$) produced by (R10–R11) can oxidize $\text{HSO}_3^-$ to sulfate (R13) with a similar rate constant to $\text{H}_2\text{O}_2 + \text{HSO}_3^-$ (Betterton and Hoffmann, 1988). The sulfate radical ($\text{SO}_4^-$) produced by (R12) is a very strong oxidant and can react rapidly with $\text{HSO}_3^-$ and $\text{SO}_3^{2-}$ (R14–R15) as well as with many other species such as $\text{Cl}^-$, $\text{NO}_2^-$, $\text{O}_2^-$, $\text{HCOO}^-$, and $\text{HO}_2$ (Jacob, 1986). The rate constants for (R10–R15) can be found in Jacob et al. (1989). It is convenient to define the sulfate yield as the number of $\text{SO}_4^{2-}$ ions produced due to each attack of $\text{OH}_{(\text{aq})}$ on HMS. If $\text{SO}_5^-$ reacts with $\text{O}_2^-/\text{HCOO}^-$ (R10–R11) and the product $\text{HSO}_5^-$ oxidizes $\text{HSO}_3^-$ (R13), the yield is 2. If $\text{SO}_5^-$ undergoes self-reaction (R12) and the produced $\text{SO}_4^-$ reacts with $\text{HSO}_3^-/\text{SO}_3^{2-}$ (R14–R15), a reaction chain is triggered as the products include $\text{SO}_5^-$. In certain conditions, the sulfate yield can reach several tens or more (Jacob et al., 1989). However, as mentioned above, other oxidizable species also compete for $\text{SO}_4^-$, thereby terminating this chain and leading to a sulfate yield of 1. In remote environments where $\text{SO}_2$ is very low, $\text{HSO}_5^-$ may be a stable species, resulting in a sulfate yield < 1. Our low $[\text{OH}]_{\text{aq}}$ assumption implies the existence of important oxidizable species, and therefore, the chain propagation is limited. As in Moch et al. (2020) the sulfate yield is assumed to be 2 in our simulations.

$$\text{SO}_5^- + \text{O}_2^- \xrightarrow{\text{H}_2\text{O}} \text{HSO}_5^- + \text{O}_2 + \text{OH}^- \tag{R10}$$

$$\text{SO}_5^- + \text{HCOO}^- \xrightarrow{\text{O}_2} \text{HSO}_5^- + \text{O}_2^- + \text{CO}_2 \tag{R11}$$

$$\text{SO}_5^- + \text{SO}_5^- \rightarrow 2\text{SO}_4^- + \text{O}_2 \tag{R12}$$

$$\text{HSO}_5^- + \text{HSO}_3^- \rightarrow 2\text{SO}_4^{2-} + 2\text{H}^+ \tag{R13}$$

$$\text{SO}_4^- + \text{HSO}_3^- \xrightarrow{\text{O}_2} \text{SO}_4^{2-} + \text{SO}_5^- + \text{H}^+ \tag{R14}$$

$$\text{SO}_4^- + \text{SO}_3^{2-} \xrightarrow{\text{O}_2} \text{SO}_4^{2-} + \text{
[revised manuscript text omitted]

**3.4 Comparison with prior observations**

The quantitative observations of HMS in ambient aerosols remain sparse (Moch et al., 2020) and we provide here a comparison between the observations at four sites and two model simulations (control and default) in Table 5. Since these observations have been collected over the past three decades while our simulations cover only one year, it is more appropriate to use the molar ratios of $HMS/SO_4^{2-}$ or HMS/MSA rather than absolute HMS concentrations. The measurement techniques of HMS in different studies have been different (Table 5) and their effects on the reported results are unclear. Moreover, observations of HMS may be subject to measurement artifacts that may make quantitative comparisons between model results and observations difficult (Ma et al., 2020; Moch et al., 2020). Among the observations shown in Table 5, the highest $HMS/SO_4^{2-}$ ratio of 11% has been found in winter in Beijing by Ma et al. (2020). Model results from both default and control simulations agree well with this observed ratio. Less HMS was observed in other regions, including New Mexico (USA), Germany, and Osaka Bay (Japan). The default simulation overestimates the $HMS/SO_4^{2-}$ or HMS/MSA ratios by a factor of 2–3, whereas the control simulation overestimates these ratios by an order of magnitude.

A more detailed comparison of the model with observations in Beijing is provided below (Ma et al., 2020). These samples from Beijing were stored at −20 °C between sample collection and analysis, had HCHO added to solution during sample extraction, and were examined by ion chromatography using a more acidic eluent than normal all in order to limit HMS decomposition and misidentification. The observations in Ma et al. (2020) cover 73 days in winter and 11 polluted days in other seasons. The data for the other seasons is presented only in their discussion paper. Because of the coarse resolution of global model, we do not expect our simulations to capture the day-to-day variability that is observed at a single site. Accordingly, we examine the ability of our simulations to reproduce the observed relationships between HMS and its influencing factors. Figure 8 provides scatter plots of HMS concentrations (and $HMS/SO_4^{2-}$ ratios) versus two variables ($O_3$ and RH) and compares the data from observations and model simulations (control and default). The level of $O_3$ represents photochemical oxidizing capacity and RH may indicate the abundance of aqueous water in the lower troposphere.

[revised manuscript text omitted]

Wang, J., Li, J., Ye, J., Zhao, J., Wu, Y., Hu, J., Liu, D., Nie, D., Shen, F., Huang, X., Huang, D. D., Ji, D., Sun, X., Xu, W., Guo, J., Song, S., Qin, Y., Liu, P., Turner, J. R., Lee, H. C., Hwang, S., Liao, H., Martin, S. T., Zhang, Q., Chen, M., Sun, Y., Ge, X., and Jacob, D. J.: Fast sulfate formation from oxidation of SO2 by NO2 and HONO observed in Beijing haze, Nature Communications, 11, 2844, 10.1038/s41467-020-16683-x, 2020.

Wei, L., Fu, P., Chen, X., An, N., Yue, S., Ren, H., Zhao, W., Xie, Q., Sun, Y., Zhu, Q.-F., Wang, Z., and Feng, Y.-Q.: Quantitative Determination of Hydroxymethanesulfonate (HMS) Using Ion Chromatography and UHPLC-LTQ-Orbitrap Mass Spectrometry: A Missing Source of Sulfur during Haze Episodes in Beijing, Environmental Science & Technology Letters, 7, 701-707, 10.1021/acs.estlett.0c00528, 2020.

[revised manuscript text omitted]

| Reaction | Rate expression (M s$^{-1}$) | Reference and note |
|---|---|---|
| $SO_2 + HCHO \rightarrow HMS$ | $R_{aq} = k_f[HCHO]_{aq}[SO_2^T]_{aq} = \left(k_1 x_{HSO_3^-} + k_2 x_{SO_3^{2-}}\right)[HCHO]_{aq}[SO_2^T]_{aq}$
 $k_{1,high} = 7.9 \times 10^2 \times \exp(-3000 \times (1/T - 1/298))$ M$^{-1}$ s$^{-1}$
 $k_{2,high} = 2.5 \times 10^7 \times \exp(-2500 \times (1/T - 1/298))$ M$^{-1}$ s$^{-1}$
 $k_{1,low} = 3.2 \times 10^2 \times \exp(-2700 \times (1/T - 1/298))$ M$^{-1}$ s$^{-1}$
 $k_{2,low} = 3.8 \times 10^6 \times \exp(-2500 \times (1/T - 1/298))$ M$^{-1}$ s$^{-1}$ | Boyce and Hoffmann (1984) |
| $HMS \rightarrow SO_2 + HCHO$ | $k_d = 6.2 \times 10^{-8} \times \exp(-11400 \times (1/T - 1/298))$
 $+4.8 \times 10^3 \times (K_w/[H^+]) \times \exp(-4700 \times (1/T - 1/298))$ s$^{-1}$ | Boyce and Hoffmann (1984); Deister et al. (1986) |
| $HMS + OH \xrightarrow{O_2} HCHO + SO_5^-$ | $k_3 = 2.7 \times 10^8$ M$^{-1}$ s$^{-1}$ | Olson and Fessenden (1992) |
| $SO_2 + O_3 \rightarrow SO_4^{2-} + O_2$ | $R_{aq} = \left(k_1 x_{SO_2 \cdot H_2O} + k_2 x_{HSO_3^-} + k_3 x_{SO_3^{2-}}\right)[O_3]_{aq}[SO_2^T]_{aq}$
 $k_1 = 2.4 \times 10^4$ M$^{-1}$ s$^{-1}$
 $k_2 = 3.7 \times 10^5 \times \exp(-5530 \times (1/T - 1/298))$ M$^{-1}$ s$^{-1}$
 $k_3 = 1.5 \times 10^9 \times \exp(-5280 \times (1/T - 1/298))$ M$^{-1}$ s$^{-1}$ | Seinfeld and Pandis (2016) |
| $SO_2 + H_2O_2 \rightarrow SO_4^{2-} + H_2O$ | $R_{aq} = k_4 K_{s1} x_{SO_2 \cdot H_2O}[H_2O_2]_{aq}[SO_2^T]_{aq}$
 For cloud water, $k_4 =$
 $7.45 \times 10^7 \times \exp(-4430 \times (1/T - 1/298))/(1 + 13[H^+])$ M$^{-2}$ s$^{-1}$
 For aerosol water, $k_4$ is multiplied by an enhancement factor EF that is dependent on $\mu_b$:
 $EF = \begin{cases} 1.5, 0 < \mu_b \leq 4 \\ 2.3\exp(2.4\log_{10}\mu_b - 1.2), \mu_b > 4 \end{cases}$ | Seinfeld and Pandis (2016); Liu et al. (2020); note $a$ |
| $SO_2 + O_2 \xrightarrow{Mn^{2+}, Fe^{3+}} SO_4^{2-}$ | $R_{aq} = k_5[H^+]^{-0.74}[Mn^{2+}][Fe^{3+}][SO_2^T]_{aq}$ (pH < 4.2)

[revised manuscript text omitted]